# Waves of actin and microtubule polymerization drive microtubule-based transport and neurite growth before single axon formation

**Amy M Winans[1,2,3], Sean R Collins[2,3†], Tobias Meyer[2,3*]**

[1]Biophysics Program, Stanford University, Stanford, United States; [2]Department of Chemical and Systems Biology, Stanford University, Stanford, United States; [3]Center for Systems Biology, Stanford University, Stanford, United States

**Abstract** Many developing neurons transition through a multi-polar state with many competing neurites before assuming a unipolar state with one axon and multiple dendrites. Hallmarks of the multi-polar state are large fluctuations in microtubule-based transport into and outgrowth of different neurites, although what drives these fluctuations remains elusive. We show that actin waves, which stochastically migrate from the cell body towards neurite tips, direct microtubule-based transport during the multi-polar state. Our data argue for a mechanical control system whereby actin waves transiently widen the neurite shaft to allow increased microtubule polymerization to direct Kinesin-based transport and create bursts of neurite extension. Actin waves also require microtubule polymerization, arguing that positive feedback links these two components. We propose that actin waves create large stochastic fluctuations in microtubule-based transport and neurite outgrowth, promoting competition between neurites as they explore the environment until sufficient external cues can direct one to become the axon.

**\*For correspondence:** tobias1@stanford.edu

**Present address:** [†]Department of Microbiology and Molecular Genetics, University of California, Davis, Davis, United States

**Competing interests:** The authors declare that no competing interests exist.

## Introduction

During development, hippocampal neurons transit through a multi-polar intermediate state in which neurons typically extend 4–5 immature neurites, which are each capable of becoming either an axon or a dendrite (*Barnes and Polleux, 2009*; *Dotti et al., 1988*). These neurites stochastically retract and elongate for a period of hours to days before a single neurite is specified as the axon (*Figure 1a*). This delayed axon specification is typically mediated by extracellular cues such as soluble growth factors, neighboring cells, and the extracellular matrix (*Arimura and Kaibuchi, 2007*; *Barnes and Polleux, 2009*) or in vitro by stochastic extension and stabilization of the longest single neurite (*Dotti et al., 1988*). Previous studies showed that axon specification is based on a competition between "axon-promoting" signals such as Ras, Phosphatidylinositol-4,5-bisphosphate 3-kinase (PI3K), and Protein Kinase A (PKA) and "axon-inhibiting" signals such as Glycogen synthase kinase 3 beta (GSK3β) and Protein Kinase G (PKG) (*Barnes and Polleux, 2009*; *Shelly et al., 2010*). It is believed that these signals are controlled by the selective accumulation of axon-promoting proteins in the nascent axon via microtubule-based transport involving one or more self-reinforcing positive feedback loops (*Figure 1a*) (*Cheng et al., 2011*; *Fivaz et al., 2008*; *Inagaki et al., 2001*; *Shi et al., 2003*; *Toriyama et al., 2006*). During the multi-polar state, however, pro-axon components appear to dynamically shuttle collectively to and from different neurites through an unknown mechanism (*Figure 1a*), a process that has been investigated using a constitutively active version of the microtubule motor Kinesin-1 (a.a. 1–560, CA-KIF5C) (*Hammond et al., 2010*; *Jacobson et al., 2006*;

**eLife digest** Nerve cells (also known as neurons) connect with each other to form complex networks through which signals are carried around the body. Signals are received by branch-like projections called dendrites, pass through the cell body and then pass along a long projection called the axon before being transmitted to the dendrites of neighboring neurons.

In animal embryos, immature neurons in part of the brain called the hippocampus – which is crucial for learning and forming memories – develop into mature neurons through a series of steps. In the early stages of development, an immature neuron sends out multiple projections that extend out in all directions from its cell body. These projections randomly retract and lengthen for a while before a single projection grows into an axon and the others become dendrites.

It is believed that signal proteins inside the neuron that promote the formation of an axon selectively accumulate in a projection as it grows into an axon. These axon-promoting proteins are carried to the axons by a motor protein called kinesin, which moves along fibers called microtubules. In immature neurons, kinesin motors randomly move in and out of different projections, before settling in the projection that will grow into the axon. However, it is not clear what drives these fluctuations.

To address this question, Winans et al. used microscopy to study the transport of axon-promoting proteins in hippocampal neurons. The experiments show that a protein called actin forms a mesh of filaments in a wave-like manner, starting in the cell body and moving outwards into the projections. When a wave of actin reaches a projection, the projection grows for a while and then stops until the next actin wave arrives. Furthermore, the actin waves promote the formation of more microtubule filaments.

This work shows that actin waves make the projections wider to create space for more microtubules to form, which increases the transport of axon-promoting proteins to the projections. Winans et al.'s findings suggest that actin waves direct axon-promoting proteins to axons and promote competition between the projections early on by generating random fluctuations that allow all the projections to grow and retract. This would allow each projection to explore its environment in the search for signals that promote axon growth. The next challenge is to understand how different signals select the "winning axon".

*Konishi and Setou, 2009*; *Toriyama et al., 2010*). Kinesin-1 carries numerous proteins known to promote axon formation such as CRMP2 (*Kimura et al., 2005*) and WAVE/Sra (*Kawano et al., 2005*), and perturbing Kinesin-1 expression or localization inhibits single axon formation (*Konishi and Setou, 2009*). There is also evidence that the Kinesin-1 adaptor protein c-Jun N-terminal kinase-interacting protein-1 (JIP1) (*Dajas-Bailador et al., 2008*) and the PI3K interactor Shootin1 (*Toriyama et al., 2006*, *2010*) also relocalizes from one neurite to another before settling in the nascent axon. A similar dynamic collective relocalization of this kinesin motor domain has also been observed in vivo in developing neurons expressing CA-KIF5C (*Randlett et al., 2011*). What drives these fluctuations is a fundamental open question, as the number, orientation, or age of microtubules may play a role, and conflicting studies suggest that Kinesin-1 may preferentially bind microtubules that are stable (*Cai et al., 2009*; *Hammond et al., 2010*; *Konishi and Setou, 2009*; *Reed et al., 2006*) or newly polymerized (*Nakata et al., 2011*; *Valesoq et al., 1994*). These results regarding microtubule-based transport may also be related to an earlier observation that bulk cytoplasmic flow precedes axon specification (*Bradke and Dotti, 1997*).

Here we show that actin waves, growth-cone-like cytoskeletal structures, promote anterograde microtubule and kinesin-based transport during the multi-polar state of symmetry breaking. Previous observations of actin waves showed that they are triggered stochastically, linked to neurite outgrowth and axon specification, more prevalent in early neurons compared to mature neurons, and found in cultured organotypic slices as well as primary hippocampal neurons (*Flynn et al., 2009*; *Katsuno et al., 2015*; *Ruthel and Banker, 1998*, *1999*). It has been suggested that they constitute a transport mechanism that brings actin and actin associated proteins towards growth cones (*Flynn et al., 2009*; *Katsuno et al., 2015*). Our study shows that actin waves act in concert with

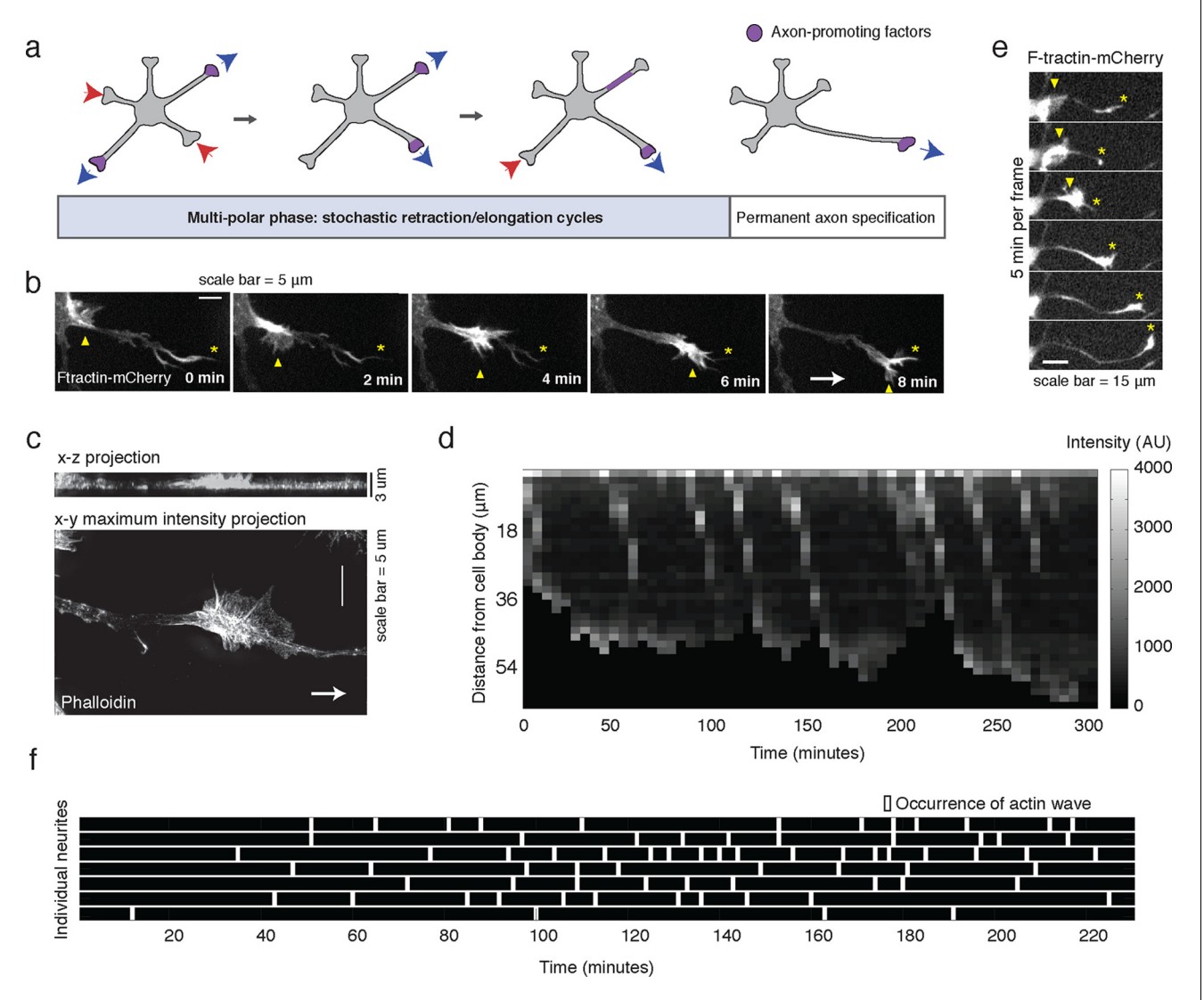

**Figure 1.** Stochastically-generated actin waves correlate with neurite extensions. (**a**) Schematic showing two stages of symmetry breaking. The multi-polar phase, where the neuron experiences fluctuating neurite outgrowth and retraction and fluctuating microtubule-based transport, is highlighted. (**b**) Timelapse images of a F-tractin-mCherry-expressing primary hippocampal neurite showing actin wave propagation. Images were taken every 2 min. Yellow arrowheads mark the actin wave, yellow asterisks mark the neurite tip, white arrow marks direction of actin wave progression. Scale bar = 5 µm. (**c**) Structured illumination images of a phalloidin-stained neuron showing an actin wave. Top image is a x-z projection of a z-stack of images taken every 0.125 µm, bottom image is a maxiumum intensity projection of the z-stack. White arrow marks direction of actin wave progression. (**d**) Kymograph generated from a timelapse of a F-tractin-mCherry expressing neurite. Source images were acquired every 5 min. (**e**) Timelapse images of a F-tractin-mCherry-expressing neurite undergoing a growth spurt as the actin wave impacts the growth cone. Images were acquired every 5 min. Yellow arrowheads mark actin waves, yellow asterisks mark neurite tips. Scale bar = 15 µm. (**f**) Actin waves are stochastically generated in different neurites over time. Actin wave generation was assessed by eye in all neurites of a single neuron over time. Horizontal bars mark individual neurites, white dashes mark actin waves. Source images were acquired every 5 min.

The following figure supplements are available for figure 1:

**Figure supplement 1.** Frequency of actin waves.

**Figure supplement 2.** Speed of actin waves.

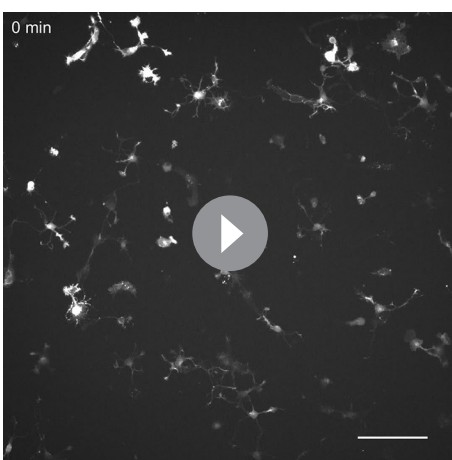

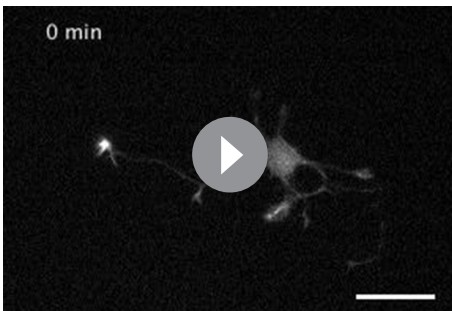

**Video 1.** Actin waves are widespread and move in an anterograde fashion. This movie shows timelapse images from the entire frame of acquisition for F-tractin-mCherry expressing neurons. Images were collected every 5 min and the movie was generated at 5 frames per second. Scale bar = 100 μm.

**Video 2.** Actin waves are generated in a seemingly stochastic fashion and move anterogradely through neurites to cause neurite extension. This movie shows timelapse images of an F-tractin-mCherry-expressing neuron producing actin waves. Images were collected every 5 min and the movie was generated at 5 frames per second. Scale bar = 30 μm.

microtubules to direct microtubule-based transport and that these discrete transport events are tightly linked to neurite outgrowth during the multi-polar state before an axon is specified.

## Results

To investigate the role of actin waves in neuronal polarity, we first characterized actin waves in our in vitro system. Consistent with previous studies of cultured hippocampal neurons extracted from embryonic rat brains (*Ruthel and Banker, 1998*, *1999*; *Flynn et al., 2009*; *Katsuno et al., 2015*) actin waves are clearly visible in all neurons and neurites that bear actively growing processes (*Video 1*), and travel nearly exclusively in an anterograde fashion, from cell body to growth cone (*Video 1*, *2*) during our DIV1 imaging window. Actin waves are generated at a median frequency of 1–2 actin waves per hour (*Figure 1—figure supplement 1*), move at average speeds of 2–3 μm/min (*Figure 1—figure supplement 2*), and morphologically resemble growth cones (*Figure 1b,c*). Structured illumination microscopy (SIM) revealed that actin waves are a mix of lamellipodial and filopodial structures that project outward from the neurite shaft, mainly contained to the x-y plane (*Figure 1c*). After an actin wave impacts the neurite tip, the neurite undergoes a growth spurt that often lasts until the growth cone shrinks (*Figure 1d, e*). Afterwards, the neurite often does not grow outwards again until the arrival of an additional actin wave reactivates the growth cone. Kymographs of actin intensity paired with neurite length over time illustrate the tight correlation between neurite outgrowth and actin wave arrival (*Figure 1d*). In fact, roughly 90% of neurite outgrowth in the DIV1 window is clearly concomitant with actin wave arrival (*Video 1*). Later stage imaging (DIV2+) displays greater neurite outgrowth in the absence of actin waves (data not shown). We further noted that actin waves entered different neurites at different times in an apparently stochastic fashion without an obvious pattern or frequency (*Figure 1f*). For the majority of live cell images in this study, fluorescently-tagged F-tractin (*Johnson and Schell, 2009*) is used as a marker for f-actin, although actin waves can also be observed with only a membrane marker (*Video 3*) as well. Together with previous findings, our results strongly support the hypothesis that actin waves have a role in promoting the stochastic fluctuations in neurite length observed during the multi-polar phase of symmetry breaking.

### Actin waves are linked to increased microtubule polymerization and increased microtubule based transport

We first hypothesized that actin waves may link to microtubules after observing an increase in neurite volume, or cross-sectional area, in and behind the actin wave (*Figure 2a*), which was confirmed with averaged line scan analysis (*Figure 2b*, *Figure 2—figure supplement 1*). Immunofluorescence

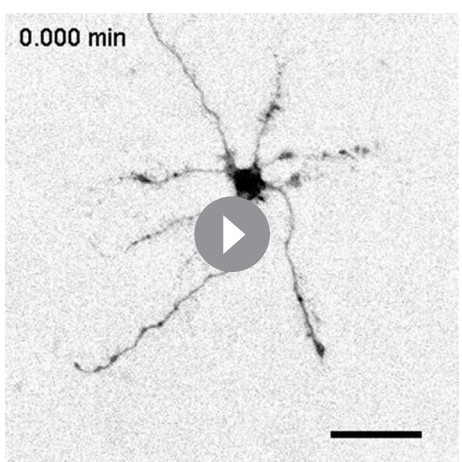

**Video 3.** Actin waves are observed in neurons expressing a membrane marker. This movie shows timelapse images of a Lyn-mCherry-expressing neuron producing actin waves. Neuron was imaged on DIV2 under CO2 in standard culturing Neurobasal Media. Images were collected every 5 min and the movie was generated at 5 frames per second. Scale bar = 40 μm.

experiments staining for neuronal tubulin revealed an increase in microtubule intensity in and behind actin waves (*Figure 2c*), which was confirmed by structured illumination microscopy (SIM) of single microtubules (*Figure 2e*) and using averaged line scans of lower-resolution immunofluorescence images (*Figure 2d*). As a control, we confirmed using lower-resolution IF (*Figure 2f*) and SIM (*Figure 2—figure supplement 2*) that the increases in tubulin levels behind versus in front of an actin wave are not the result of a gradual thickening of neurite shafts closer to the cell body (microtubule number ratio in *Figure 2—figure supplement 2* is statistically greater than control measurements in *Figure 2f*).

We then set out to find the source of the increased microtubule number in and behind actin waves and tested the hypothesis that actin waves contain more polymerizing microtubules by imaging neurons that co-express F-tractin and fluorescently-tagged EB1, which binds to the plus ends of growing microtubules. Indeed, live cell imaging of F-tractin and EB1 revealed an increase in the number of EB1 puncta within and behind actin waves, with much lower levels of EB1 puncta in front of the wave (*Figure 2g*, *Figure 9b*). This enrichment in the number of polymerizing microtubules progresses down the neurite with the actin wave, creating a dual wave of polymerizing microtubules and polymerizing actin. Control experiments using a maximum intensity projection of time lapse images showed that single EB1 puncta moved in a persistent fashion outward, confirming that they track with anterograde polarized polymerizing microtubules (*Figure 2—figure supplement 3*), consistent with previous measurements showing that microtubules generally polymerize outward during the multi-polar phase (*Stepanova et al., 2003*).

This striking co-localization of polymerizing actin and increased microtubule polymerization prompted us to investigate the correlation between actin waves and microtubule-based transport by using the minimal Kinesin 1 motor domain construct (CA-KIF5C-Venus; a.a. 1–560, adapted from *Jacobson et al. (2006)*). The attachment and movement of this motor domain to microtubules is not regulated by conformational changes and is constitutively active (*Friedman and Vale, 1999*). The mechanisms controlling the dynamic localization pattern of CA-KIF5C during symmetry breaking have previously been investigated, and the dynamic, stochastic nature of actin wave entry into various neurites struck us as reminiscent of the dynamic localization of CA-KIF5C. We first confirmed that the Kinesin-1 motor, CA-KIF5C, is often enriched at the tip of one or sometimes more neurites (*Jacobson et al., 2006*) during the multipolar phase (*Figure 3a*). We also confirmed dynamic switching of the localization of CA-KIF5C back and forth between one or more neurites via the cell body over a period of many hours until most of the construct ultimately localizes to the emerging axon (*Video 4*).

Next, we determined whether the stochastic appearance of actin waves in neurites is temporally correlated with that of CA-KIF5C. We marked the time when actin waves entered neurites by visual inspection and averaged the time course of change in total CA-KIF5C intensity in the neurite receiving the actin wave. This analysis showed in the same neurite a marked increase in CA-KIF5C over a period of over 10 min following an actin wave (*Figure 3b*). Moreover, we found that when CA-KIF5C entered a neurite previously lacking CA-KIF5C, such entry events occurred in parallel with an actin wave (in 34 out of 35 entry events). We then assessed the time lag (if any) between the increases of actin and CA-KIF5C signals during CA-KIF5C entry events and found that the increase in actin intensity preceded the increase in CA-KIF5C intensity by several minutes (*Figure 3c*). To address the concern that the co-localization of actin waves and CA-KIF5C entry

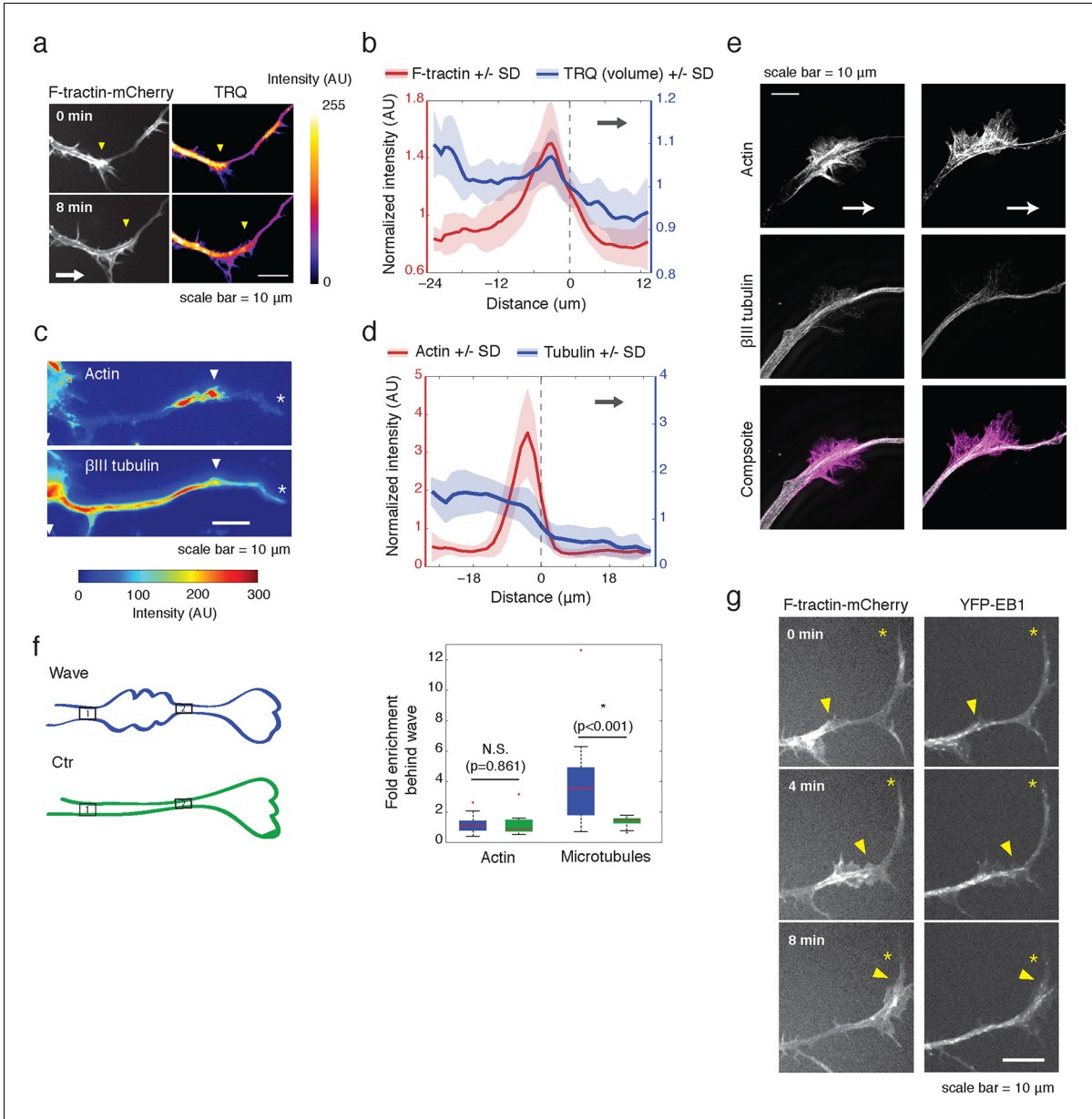

**Figure 2.** Actin waves contain more polymerizing microtubules in widened neurites. (a) The volume marker cytoplasmic Turquoise shows an increase in volume in and behind the wave. Images acquired every 8 min. Yellow arrowheads mark actin waves. White arrow denotes direction of wave movement. Scale bar = 10 μm. (b) Averaged line scans show increased volume in and behind actin wave. Measurements are taken from cytoplasmic Turquoise and F-tractin-mCherry expressing cells. Gray arrow denotes direction of wave movement. Dashed line indicates alignment at half max of actin wave. All traces were normalized by mean intensity then smoothed before averaging. Error is standard deviation. N = 14 neurites. (c) Fixed hippocampal neurons stained with phalloidin (actin) and anti-βIII tubulin (neuronal microtubules) show enrichment of microtubules in and behind wave. White arrowheads mark actin waves, white asterisks mark neurite tips. Scale bar = 10 μm. (d) Quantification of (c) confirmed enrichment of microtubules in and behind wave. Averaged line scans of phalloidin signal and anti-βIII tubulin signal were obtained for neurites containing waves. See 2b for methodology. N = 27 neurites. (e) Structured illumination microscopy on phalloidin and anti-βIII tubulin-stained neurons shows enrichment of microtubules behind wave with single-microtubule resolution. White arrow marks direction of actin wave propagation. Scale bar = 10 μm. (f) Fold enrichment of phalloidin and anti-βIII tubulin intensity behind the wave was calculated by taking the ratio of intensities in an area behind the actin wave to an area in front of the actin wave (depicted on left: region1/region2). Fold enrichment was calculated for neurites containing waves ("Wave", n = 20) and neurites lacking waves ("Ctr", n = 12). Tubulin enrichment was statistically higher in waves compared to the control (two-sided Wilcoxon rank sum test). (g) Hippocampal neurons expressing F-tractin-mCherry and YFP-EB1 show enrichment of EB1 puncta in and behind actin wave. Yellow arrowheads mark front edge of actin waves. Yellow asterisks mark neurite tips. Scale bar = 10 μm.

The following figure supplements are available for figure 2:

*Figure 2 continued on next page*

*Figure 2 continued*

**Figure supplement 1.** 2D line scan analysis method.
**Figure supplement 2.** Single microtubule enrichment behind actin wave.
**Figure supplement 3.** EB1 puncta move in an anterograde fashion.

into neurites may be an artifact of a cytoplasmic fraction of CA-KIF5C, we imaged CA-KIF5C moving in and out of neurites in neurons also expressing a cytoplasmic Turquoise fluorescent protein. Taking the ratio of CA-KIF5C over the Turquoise volume marker shows that the movement of CA-KIF5C into and out of neurites is significant over changes in volume; moreover, addition of the microtubule polymerization inhibitor Nocodazole eliminates the movement of CA-KIF5C into and out of neurites relative to a volume marker, confirming that CA-KIF5C movement is dependent on microtubules (*Figure 3—figure supplement 1*).

In a parallel analysis of the spatial correlation between actin polymerization and CA-KIF5C, we found that CA-KIF5C was transported in pulses, or waves, along the neurite shaft to the growth cone, coincident with actin waves (*Figure 3d*, *Figure 3—figure supplement 2*, *Video 5*). We confirmed this correlation using single cell analysis (*Figure 3e*), averaged line traces (*Figure 3f*), and kymographs (*Figure 3g*). Most of the pulses of CA-KIF5C arriving at growth cones resulted in an increase in CA-KIF5C concentration that persisted for tens of minutes (*Figure 3g*). Moreover, the arrival of an actin wave can reverse the retrograde movement of CA-KIF5C out of the neurite (*Figure 3—figure supplement 3*). Taken as a whole, this data shows a strong connection between CA-KIF5C movement and actin waves.

To further verify an increase in microtubule-based transport in actin waves, we expressed fluorescently-tagged Synaptophysin, a pre-synaptic vesicle marker, in order to image vesicular movement. Similar to CA-KIF5C localization, Synaptophysin-positive vesicles were highly enriched in and behind actin waves (*Figure 4a*). Averaged line scans of intensity profiles of only moving vesicles (Moving Vesicle Intensity (MVI)) showed the same results (*Figure 4b* see Materials and methods). Higher frequency imaging showed that Synaptophysin-positive vesicles enriched in the actin waves were not subject to the Brownian motion suggesting that they remain attached to microtubules (*Video 6*). Taken as a whole, this data suggests a strong correlation between actin waves, increased microtubule polymerization, and increased microtubule based transport.

## Actin waves may spatially de-restrict microtubule polymerization to increase microtubule-based transport

We next examined whether the striking connection between actin waves, microtubule polymerization and microtubule-based transport are causal. We first probed the potential role of actin waves in promoting microtubule polymerization and microtubule-based transport by examining both microtubule polymerization and microtubule-based transport after the addition of Jasplakinolide, an actin-stabilizing agent that stalls the progression of actin waves. We found that stalling actin waves halted the progression of CA-KIF5C (*Figure 5a,b*) as well as the progression of the wave of polymerizing microtubules (*Figure 5c,d*), suggesting that progression of actin waves was necessary in order to continue the forward progression of the other two components. Notably, EB1 comets were still observed in the stalled actin wave, suggesting that although microtubules were still polymerizing in the actin waves, they were unable to polymerize beyond the structural bottleneck present at the front of the actin wave. This observation suggests that a mechanical mechanism may, at least partially, drive the increase in microtubule polymerization observed in the actin wave, and may explain the conundrum presented by the discrepant speeds of the components moving along with actin waves. More precisely, vesicular transport and Kinesin motors both advance at a speed of ~50 μm/min along microtubules while actin waves only advance at a speed of ~ 2 μm/min (*Figure 1—figure supplement 2*). This raises important mechanistic questions of why Kinesin motors and vesicles have a net progression that is much slower than their molecular transport rate, why motors and vesicles appear to be restrained behind the leading edge of actin waves, and why vesicular transport stays elevated in and behind an actin wave (*Figure 4a*).

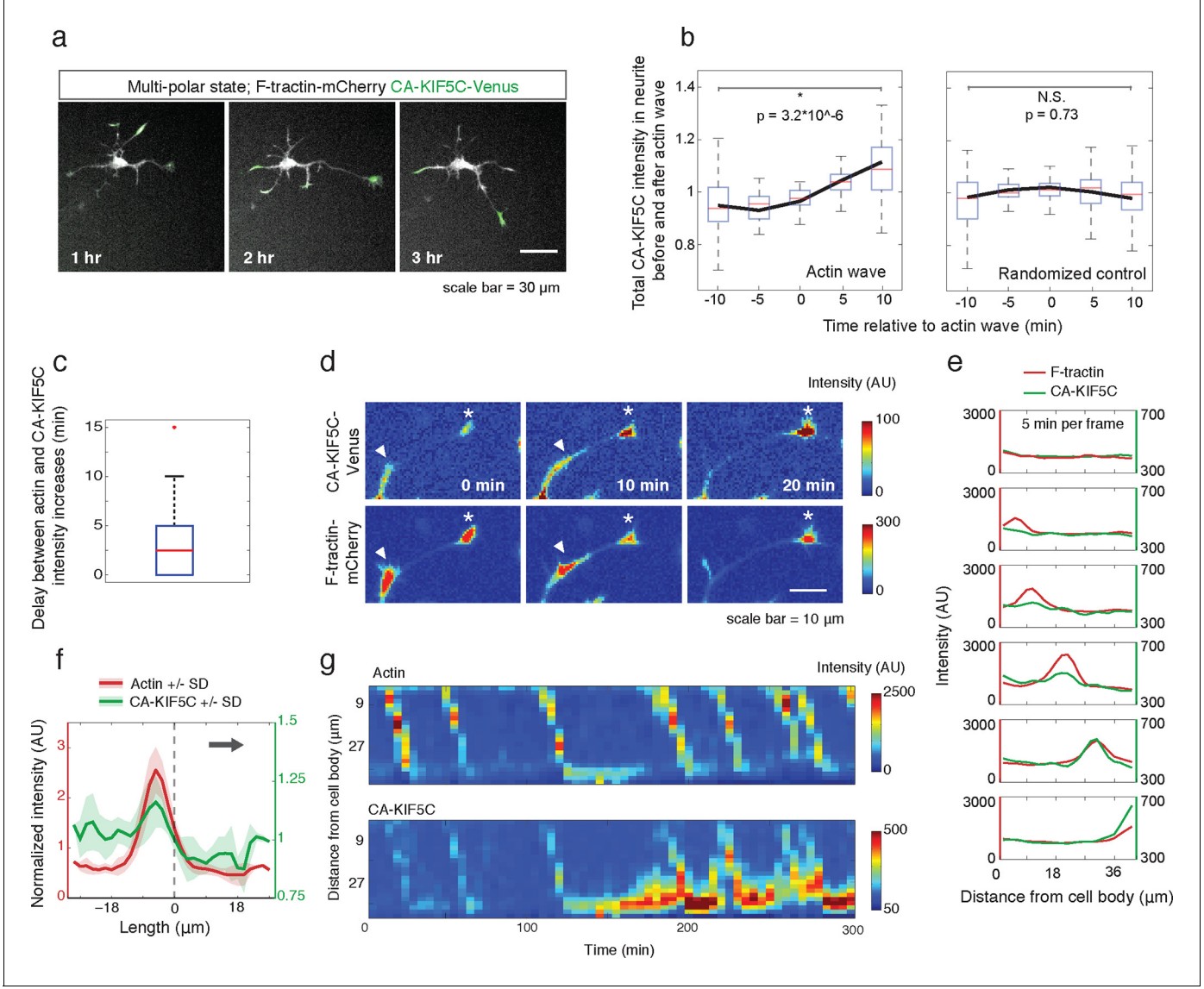

**Figure 3.** Actin waves coordinate with pulsatile transport of Kinesin-1 motor domain. (**a**) Live cell images of a neuron expressing CA-KIF5C-Venus (green) and F-tractin-mCherry (white) exhibiting fluctuating CA-KIF5C localization and neurite lengths characteristic of the multi-polar stage. Images were acquired every hour. Scale bar = 30 μm. (**b**) For 45 visually-identified actin waves, total CA-KIF5C intensity in the neurite was measured before and after generation of the actin wave. A significant increase in CA-KIF5C is observed relative to a control. Control traces were obtained by randomly selecting 45 points in time and assessing CA-KIF5C intensity before and after each time point. The black line signifies the mean of the CA-KIF5C traces. For each time point the data is also represented with standard box plots with outliers not shown. Significance between the -10 and 10 min set of points was assessed using a two-sided Wilconox rank-sum test. (**c**) Increase in actin intensity precedes increase in CA-KIF5C intensity. Total intensities of actin and CA-KIF5C before, during, and after entry of CA-KIF5C was assessed and the delay between actin and CA-KIF5C intensity increase was noted. (28 entry events). Source images were acquired every 5 min. (**d**) Timelapse images of a CA-KIF5C-Venus and F-tractin-mCherry expressing neurite show that CA-KIF5C transports in pulses which coincide with actin waves. Images were taken every 10 min. White arrowheads mark position of actin waves. White asterisks mark neurite tips. Scale bar = 10 μm. (**e**) CA-KIF5C moves with an actin wave as illustrated by single cell successive line scans taken from a neurite with a traveling actin wave. Image data is in *Figure 3—figure supplement 2*. Frames were acquired every 5 min. (**f**) Averaged line scans show enrichment of CA-KIF5C in and behind the actin wave. See 2b for methodology. (**g**) Kymographs generated from timelapse images of a F-tractin-mCherry and CA-KIF5C-Venus expressing neurite show that CA-KIF5C travels with actin waves and accumulates in the growth cone. Source images were acquired every 5 min.

The following figure supplements are available for figure 3:

**Figure supplement 1.** CA-KIF5C dynamic localization is dependent on microtubules.

*Figure 3 continued*

**Figure supplement 2.** CA-KIF5C travels with actin waves.

**Figure supplement 3.** Actin waves can reverse retrograde CA-KIF5C movement.

To answer these discrepancies, we speculated that the bottleneck at the front of the actin wave restrains the movement of polymerizing microtubules and microtubule-based transport, and that the dilation of the neurite shaft caused by the actin wave may spatially de-restrict microtubules and allow more room for microtubules to polymerize towards the growth cone. In this hypothesis, the increase in the number of microtubules could enhance the flux of microtubule-based transport through the neurite. Another motivation for this hypothesis was that the speed of microtubule polymerization is ~10x faster than the speed of actin waves (in neurite: 20.4 +/- 5.4 um/min (39 measurements) in wave: 16.8 +/- 5.4 (23 measurements)), which is consistent with previously measured microtubule polymerization rates (*Stepanova et al., 2003*). Thus, in order for the wave of EB1 puncta to advance at the same speed as the actin wave, individual EB1 puncta must stall and disappear at the leading edge of the actin wave, similar to what we see when a polymerizing microtubule hits the membrane of the neurite tip (*Figure 6a*, left). Indeed, kymograph analysis of EB1 comets moving through the bottleneck of the actin wave revealed that EB1 comets disappear at the bottleneck (*Figure 6a*, right). As a separate experiment, we quantifiedthe flow of EB1 puncta through two windows placed right before and right after the bottleneck (but within 6 µm of each other) at time points 100 s apart which showed the same marked difference consistent with a loss of EB1 comets in between. In control experiments, we observed that the flow through each window does not significantly change over the time window measured, suggesting that microtubule polymerization is at a steady state over short time periods (*Figure 6b*). Also, the difference between the flow of EB1 puncta through the two windows does not change over time (*Figure 6—figure supplement 1*). Thus, the larger flow of polymerizing microtubules through the window closer to the cell body does not reach the second window ahead of the actin wave, suggesting EB1 puncta disappear between the two windows. To understand this result, it is helpful to again consider that processive EB1 puncta should move 25–30 µm in approximately 100 s, much faster than the actin waves. We also measured EB1 puncta flow versus neurite width in a smooth neurite or within an actin wave which reveals a correlation between neurite width and the number of EB1 puncta (*Figure 6c*). The lack of a pronounced higher number

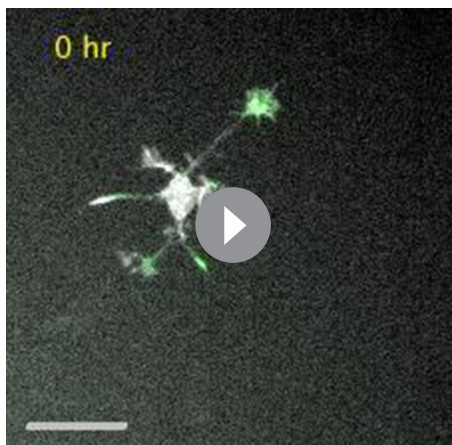

**Video 4.** CA-KIF5C switches between neurites before localizing into a single neurite. This movie shows timelapse images from a neuron expressing F-tractin-mCherry (white) and CA-KIF5C-Venus (green). Images were collected every 15 min and the movie was generated at 5 frames per second. Scale bar = 30 µm.

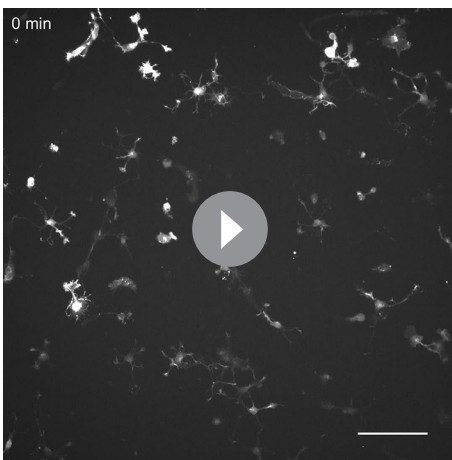

**Video 5.** Pulsatile CA-KIF5C transport coincides with moving actin waves. This movie shows timelapse images of the F-tractin-mCherry- and CA-KIF5C-Venus-expressing neurons displayed in *Figure 3d*. Images were collected every 5 min and the movie was generated at 5 frames per second. Scale bar = 30 µm.

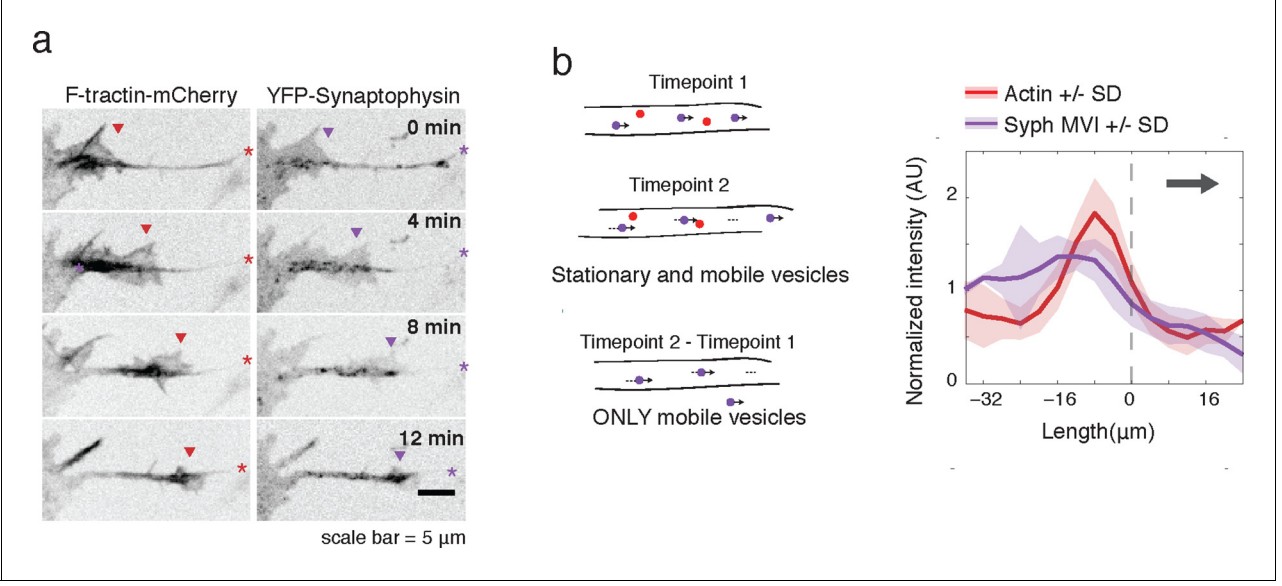

**Figure 4.** Actin waves contain Synaptophysin-positive vesicles. (a) Timelapse imaging of a neurite expressing F-tractin-mCherry and Citrine-Synaptophysin shows Synaptophysin positive vesicles enriched in and behind wave. Images displayed with inverted grayscale. Frames were taken every 4 min. Red and purple arrowheads mark front edge of actin waves. Red and purple asterisks mark neurite tips. Half asterisks mark neurites continuing out of frame. Scale bar = 5 µm. (b) Difference imaging of images acquired every 600 ms (schema left) taken as average line scans shows enrichment of mobile vesicles in and behind actin waves (right). Grey arrow denotes direction of wave movement. Dashed line indicates alignment at half max of actin wave. All traces were normalized by mean intensity then smoothed before averaging. Error is standard deviation. N = 12 neurites.

of EB1 puncta per unit area in actin waves relative to smooth neurites suggests that the mechanism by which actin waves promote microtubule polymerization is mainly steric, although this does not rule out other signaling cross-talks between the two cytoskeletal components. Furthermore, as for microtubule polymerization, the number of microtubules present also correlates with neurite width, as shown by SIM (*Figure 6d*).

Consistent with an expected role for Rac-regulated actin polymerization in generating actin waves, we used a Raichu FRET reporter (*Komatsu et al., 2011*) and found high Rac activity in and around actin waves. Interestingly, we also used a Cdc42 Raichu FRET reporter and observed that only Cdc42 exhibited a relatively higher activity in front of the actin wave, suggesting that the consistent anterograde direction of actin waves propagation may in part be directed by Cdc42 (*Figure 7a,b*). To directly investigate the dependence of microtubule polymerization on actin waves, we used a photo-activatable Rac1 construct designed by the Hahn laboratory to test whether Rac activity is sufficient to generate actin waves (*Wu et al., 2009*). This approach led to the successful generation of either fully processive (5 times) or partially processive (5 times) actin waves during successful Rac1 activations (18 times) using selective photo-excitation at the base of a neurite on a scanning confocal microscope (example in *Figure 7c*). The generated waves caused neurite widening (*Figure 7—figure supplement 1*) and increased numbers EB1 puncta (*Figure 7d*), consist with naturally-generated waves. This suggests that Rac1 activity is sufficient to initiate actin waves and to generate the enrichment of microtubule polymerization observed in actin wave.

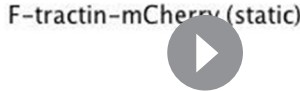

**Video 6.** Synaptophysin-positive vesicles in actin wave do not experience Brownian motion. This movie shows timelapse images of an F-tractin-mCherry and Citrine-Synaptophysin expressing neurite with an actin wave. A single F-tractin-mCherry image was taken to identify the actin wave, followed by timelapse imaging of Citrine-Synaptophysin. Images were acquired every 600 ms and the movie was generated at 5 frames per sec. Scale bar = 10 µm.

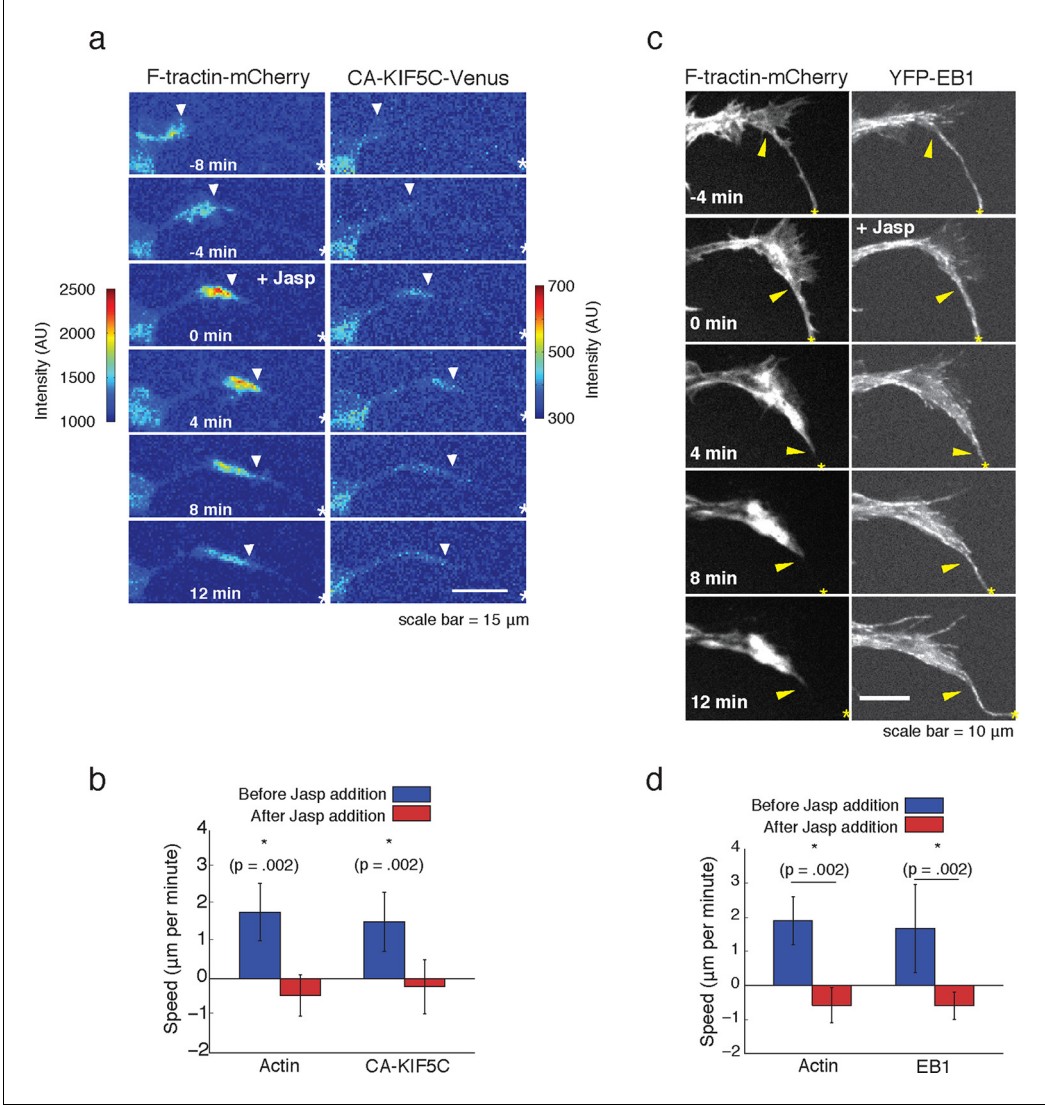

**Figure 5.** Forward advance of microtubule polymerization and Kinesin-1 is dependent on actin wave progression. (a) Wave of CA-KIF5C does not advance independently of actin wave advancement. Addition of 10 nM Jasplakinolide stalls actin wave and movement of CA-KIF5C. Frames were acquired every 4 mins. White arrowheads mark actin waves. White half asterisks mark neurites continuing out of frame. Scale bar = 15 μm. (b) Quantification of (a). Speeds of actin waves and CA-KIF5C waves were measured before and after Jasplakinolide addition. Error bars represent standard deviation. N = 6 neurites. Statistical significance assessed with a two-sided Wilcoxon rank-sum test. (c) Wave of polymerizing microtubules does not advance independently of actin wave. Addition of 50 nM Jasplakinolide freezes actin wave and prevents wave of EB1 puncta from moving forward. Images were taken every 4 min. Yellow arrowheads mark front edge of actin waves. Yellow asterisks mark neurite tips. Half asterisks mark neurites continuing out of frame. Scale bar = 10 μm. (d) Quantification of (c). Speeds of actin waves and waves of EB1 puncta were measured before and after Jasplakinolide addition. Error bars representation standard deviation. N = 6 neurites. Statistical significance assess with a two-sided Wilcoxon rank-sum test.

## Positive feedback links actin waves and microtubules

Our data thus far has shown that the increases in microtubule polymerization is dependent on actin waves. We next investigated the corollary question of whether microtubules may also have a role in the advancement of actin waves. Indeed, consistent with a previous study (*Ruthel and Banker, 1998*), we found that the addition of Nocodazole caused actin waves to dissolve (*Figure 8a, b*). We also found that the addition of a high dose of Taxol (50 nM), a small molecule that stabilizes

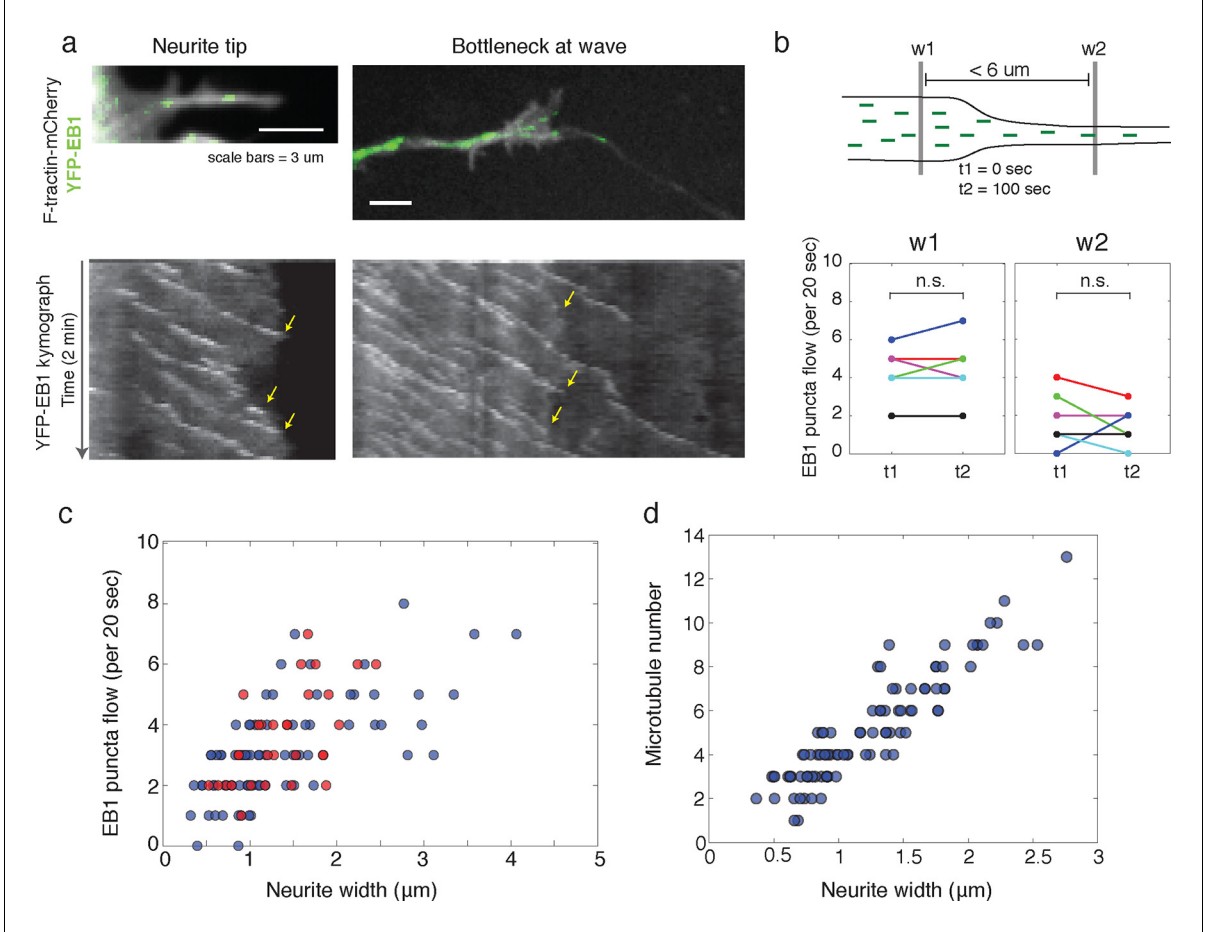

**Figure 6.** Structural bottleneck provided by actin waves inhibits progression of polymerizing microtubules. (a) Image depicting EB1 puncta at a neurite tip (top left) and at a bottleneck provided by an actin wave (top right). Accompanying kymographs illustrating EB1 puncta disappearing at the growth cone tip (bottom left) and the bottleneck of an actin wave (bottom right) are below. For better signal, the actin image of the neurite tip was constructed using a maxiumum intensity projection. Images were acquired every 2 sec for 2 min. Scale bars are both 3 µm, with the horizontal axis of the kymographs matching the spatial scale of the images. (b) Flow of EB1 puncta is restricted by a bottleneck. Flow (number of EB1 puncta through a plane over 20 sec) was assessed in two windows <6 µm apart (w1 and w2) on each side of a structural bottleneck at two time points separated by 100 sec. Flow through each window does not significantly increase or decrease between t1 and t2, however the number of EB1 puncta moving through w1 was significantly higher than the number moving through w2 at each time point. Different colors denote different neurites. Signicance was assessed using a two-sided sign test (testing difference between measurement in t1 and t2 (p = 1 for w1 and p = 0.6 for w2) and between w1 and w2 (p = 0.03, for both t1 and t2)). Colors represent distinct neurites. (c) Flow of EB1 puncta (defined in (b)) assessed in neurites with waves (red) and without waves (blue) of varying widths. Both sets (red and blue) display a linear correlation between neurite width and puncta flow. Flows per unit width for neurites bearing waves falls within the distribution for neurites lacking waves. Pearson's correlation coefficients are 0.60 for wave case and 0.68 for smooth neurite case. (d) Analysis of individual number of microtubules (visualized with SIM) assessed in neurites of varying widths shows a positive correlation. Measurements made in 9 distinct neurites.

The following figure supplements are available for figure 6:

**Figure supplement 1.** The difference in flow between windows 1 and 2 does not change over time.

**Figure supplement 2.** LatA treatment can cause neurite widening.

microtubules, leads to non-processive bursts of actin polymerization, interfering with normal actin wave formation (*Figure 8c*). 50 nM Taxol also appears to affect microtubule polymerization (*Figure 8—figure supplement 1*), indicating that microtubule stabilization and/or an effect on microtubule polymerization may be affecting actin wave production. These drug studies argue for a co-dependence between actin waves and microtubules – one is needed to advance the other.

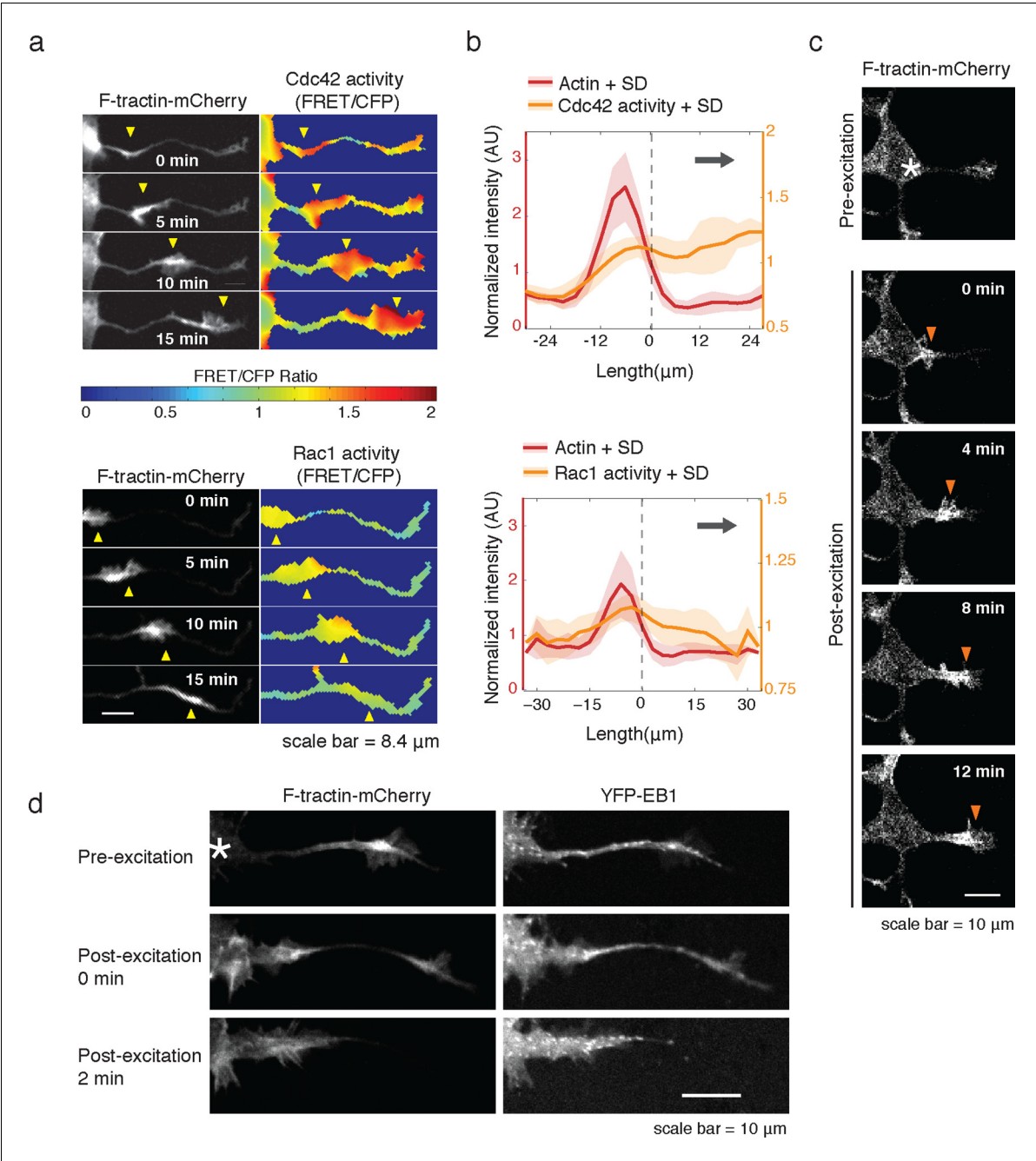

**Figure 7.** Rac1 activity is sufficient to generate actin waves enriched in polymerizing microtubules. (a) Actin waves are high in Cdc42 and Rac1 activity. Neurons are expressing F-tractin-mCherry (left) and FRET sensors for Cdc42 (top) and Rac1 activity (bottom), images were taken every 5 min. Scale bar = 8.4 μm. (b) Averaged line scans show enrichment of Cdc42 activity in and in front of the wave (top) and enrichment of Rac1 activity in the wave (bottom). Methodology in 2b. N = 22 neurites (Cdc42), n = 19 neurites (Rac1). (c) Neuron expressing F-tractin-mCherry and Cerulean-PA-Rac1 generates actin wave upon local excitation. White asterisk marks excitation area. Excitation protocol is described in Materials and Methods. Images were acquired every 4 min. Scale bar = 10 μm. (d) Neuron expressing F-tractin-mCherry, Cerulean-PA-Rac1 and YFP-EB1 shows stereotypical widening and increase in EB1 puncta upon excitation of actin wave. White asterisk marks excitation area. Scale bar = 10 μm.

The following figure supplement is available for figure 7:

**Figure supplement 1.** Activation of PA-Rac1 leads to neurite widening.

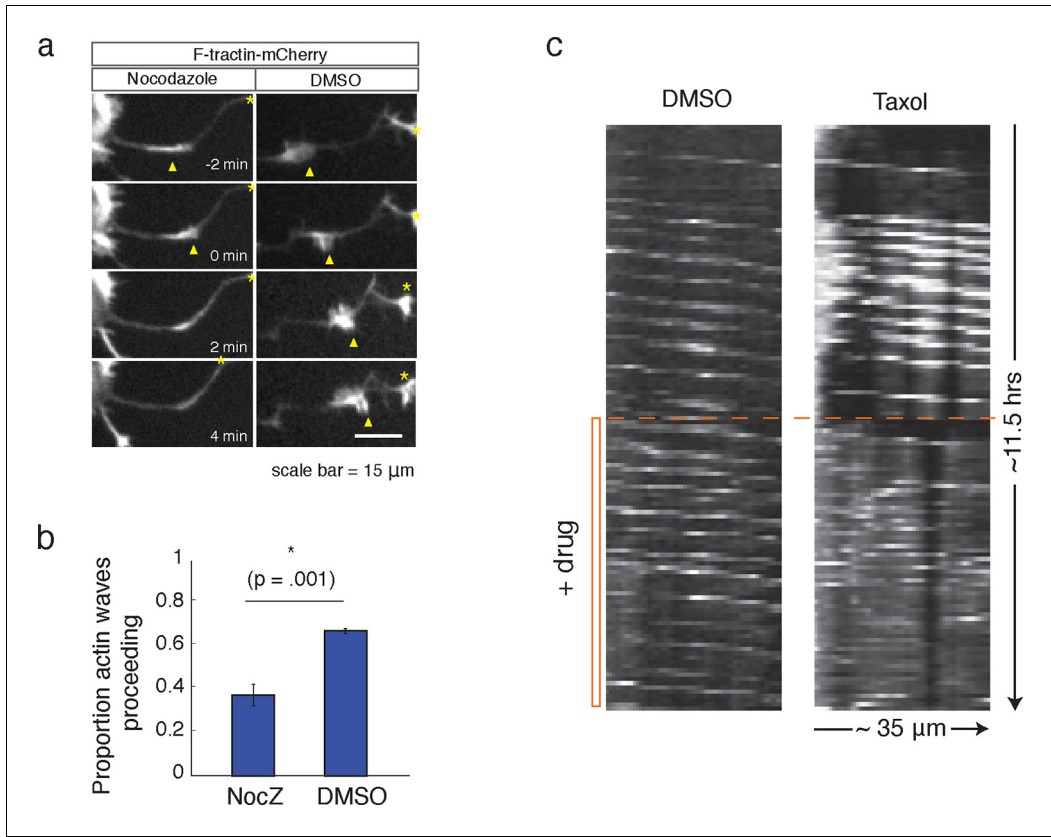

**Figure 8.** Microtubules are necessary to drive actin wave progression. (**a**) Actin waves are dependent on polymerizing microtubules. Addition of 1 μM Nocodazole dissolves an actin wave. Frames were taken every 2 min. Yellow arrowheads mark front edge of actin waves. Yellow asterisks mark neurite tips. Half asterisks mark neurites continuing out of frame. Scale bar = 15 μm. (**b**) Quantification of (**a**), showing that addition of Nocodazole dissolves a significantly greater proportion of actin waves than addition of a control. Data averaged from 3 experiments. Statistical significance assessed with 2 sample t-test. (**c**) Addition of 50 nM Taxol causes non-processive actin polymerization. Neurons were imaged for 69 frames before addition of DMSO or Taxol, then imaged for another 69 frames (5 min per frame). Neurons were expressing F-tractin-Citrine. Kymographs were constructed in Fiji.

The following figure supplement is available for figure 8:

**Figure supplement 1.** 50 nM Taxol can affect microtubule polymerization.

## Changes in microtubule polymerization and microtubule-based transport are transient

Finally, we measured how long the effect of enhanced microtubule polymerization and microtubule-based transport lasts. If such changes were long-lasting, we would expect that neurites would steadily thicken over time, which has not been previously observed. Also, our analysis of neurite growth over 10 hr time periods suggest that the growth promoting effect of an incoming actin waves is lost after tens of minutes (*Figure 1d*). To determine whether the induced microtubule polymerization persists after an actin wave, we imaged the number of EB1 puncta in a neurite segment before, during, and after actin waves (*Figure 9a*, top). Markedly, the EB1 puncta number increased along with an actin wave and persisted after the wave passed (*Figure 9a*, bottom; *Figure 9b*), in agreement with our analysis of the persistence of the spatial profiles of Synaptophysin, volume, and microtubule enrichment. We then generated kymographs of actin intensity, EB1 intensity, and neurite width in a single section of neurite with multiple actin waves (*Figure 9c*). The kymographs reveal simultaneous increases in EB1 intensity and neurite width that remained past the passage of the

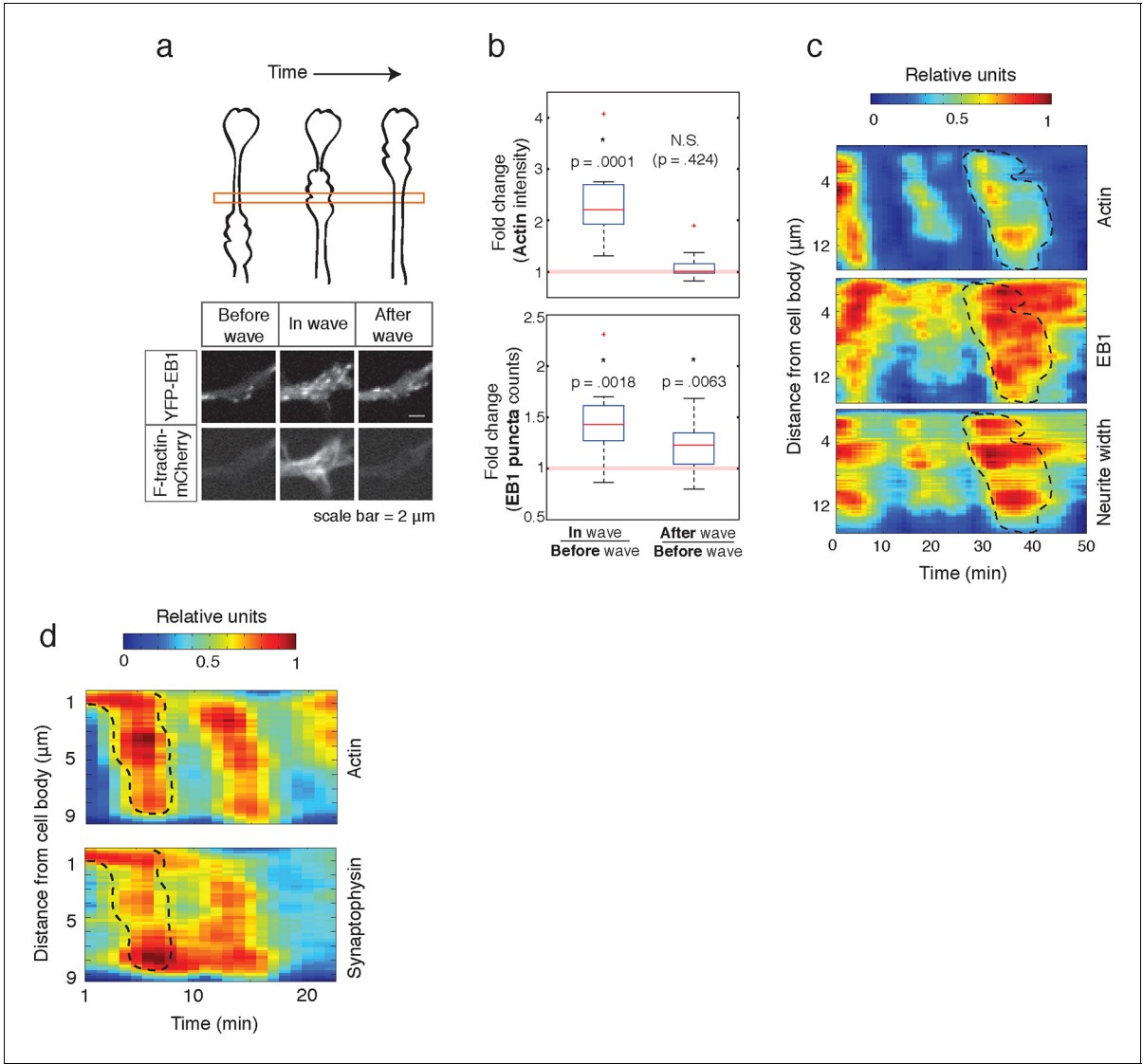

**Figure 9.** Individual actin waves drive transient increases in neurite width, microtubule polymerization, and microtubule-based transport. (**a**) EB1 puncta observed in a single section of neurite shaft before, during, and after wave progression (schema, top). Images show lingering EB1 puncta after wave has passed (bottom). Scale bar = 2 μm. (**b**) Quantification of (**a**). Fold change of EB1 puncta counts and actin intensity in the wave versus before the wave, and after the wave versus before the wave, show increased numbers of EB1 puncta in the wave and lingering enrichment of EB1 puncta after the wave has passed. N = 14, a two-sided sign test was used to asses statistical significance of a set of ratios distinct from 1. (**c**) 2D kymograph were generated from timelapse images of hippocampal neurons expressing F-tractin-mCherry and YFP-EB1 (*Video 7*). Width was calculated from segmenting summed actin and EB1 image. Region containing actin wave was marked with a dashed line and superimposed on width and EB1 kymographs. Each kymograph was normalized from 0 to 1. (**d**) 2D Kymograph shows transient enrichment of Synaptophysin vesicles in actin waves. Kymograph generated from timelapse imaging data (*Video 8*). Region containing an actin wave was marked with a dashed line and superimposed on the Synaptophysin kymograph. Each kymograph was normalized from 0 to 1.

actin wave, but gradually decayed with variable timescales (*Figure 9c*, *Video 7*). The transient nature of the increases in microtubule polymerization and microtubule-based transport was confirmed using a kymograph analysis of Synaptophysin vesicles in a neurite containing actin waves (*Figure 9d*, *Video 8*). Thus, our data as a whole suggests that each actin wave, by creating a transient increase in neurite width, creates a burst of microtubule polymerization and microtubule-based transport that will eventually decay after approximately 30 min and neurites will start to retract (*Figure 1d*, *Figure 10a*). The transient characteristic of the increase in transport explains the frequent neurite retractions and the necessity of frequent actin waves to continually deliver cargo to a single growth

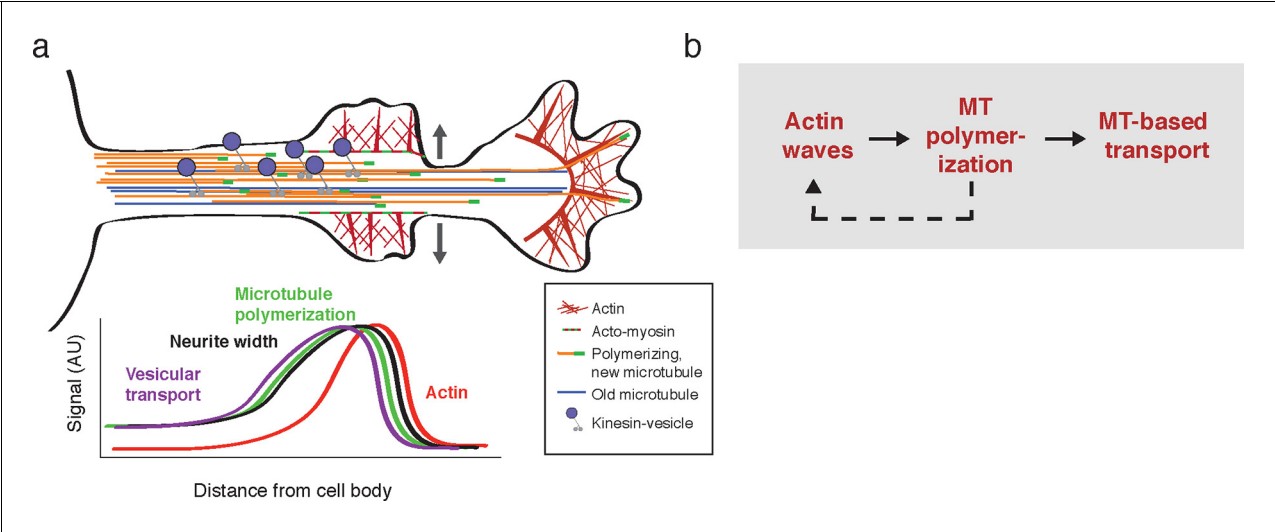

**Figure 10.** Model illustrating the co-regulation of actin and microtubules that drive microtubule-based transport. (**a**) Schematic of changes caused by the actin wave. Actin waves cause transient increases in cargo delivery by increasing microtubule polymerization. We propose that actin waves aid microtubule polymerization by widening the neurite, allowing more space for microtubules to polymerize within the shaft and leading to an increase in microtubule-based transport. However, the changes are transient and will fade in time. (**b**) Flow chart depicts working model: positive feedback between actin waves and microtubule polymerization increases microtubule-based transport.

cone to maintain and elongate the neurite and ultimately allow one of the neurites to dominate and become the axon.

## Discussion

Our study argues that developing neurons employ an interlinked cytoskeletal system whereby actin waves cross-talk with microtubules to direct microtubule-based transport and drive neurite extension. More broadly, our study suggests that the stochastic nature of actin waves leads to the stochastic increase of microtubule-based transport paired with growth cone extension in different neurites thereby creating the dynamic multi-polar state that allows a search for external cues and ultimately enables single axon formation. In particular, we show that the fluctuating actin waves control the previously observed pulsatile anterograde Kinesin transport generated during the multipolar phase of symmetry breaking. We discovered that the link from actin wave to Kinesin-mediated transport appears to be based on a

**Video 7.** Actin waves coincide with transiently increased microtubule polymerization and neurite width. This movie shows timelapse images of an F-tractin-mCherry and YFP-EB1 expressing neuron generating actin waves. As actin waves moves through, increases in neurite width and EB1 puncta number were observed. Spatially cropped images were used to generate *Figure 9c* kymograph. Images were acquired every min and the movie was generated at 5 frames per sec. Scale bar = 8 μm.

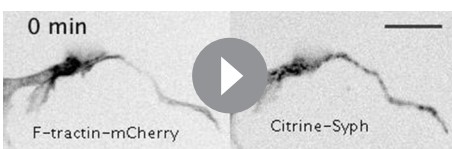

**Video 8.** Actin waves coincide with transiently increased numbers of Synaptophysin-positive vesicles. This movie shows timelapse images of an F-tractin-mCherry and Citrine-Synaptophysin expressing neuron generating actin waves. Actin waves coincide with increased numbers of vesicles. Spatially cropped images were used to generate *Figure 9d* kymograph. Images were acquired every min and the movie was generated at 5 frames per sec. Scale bar = 10 μm.

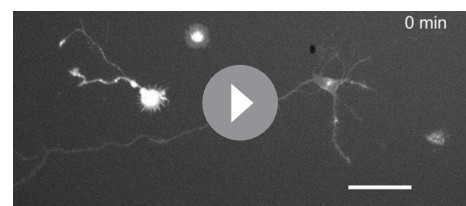

**Video 9.** Actin waves are observed in post-polarized neurons. This movie shows timelapse images of a polarized Lyn-mCherry-expressing neuron producing actin waves. Neuron was imaged on DIV2 under CO2 in standard culturing Neurobasal Media. Images were collected every 5 min and the movie was generated at 5 frames per second. Scale bar = 50 μm.

mechanical control mechanism whereby actin waves widen the neurite shaft and create the space needed for more microtubules to polymerize. The increase in microtubule polymerization in turn increases microtubule number and vesicular microtubule-based transport. In addition, these morphological changes and consequent changes in microtubule-based transport are transient, resulting in the pulsatile delivery of cargo to the growth cone and necessitating the generation of frequent actin waves in order to continue to deliver axon-promoting factors into the neurite. Finally, the results of our pharmacological perturbation of microtubules suggest that, in fact, actin waves and microtubules co-regulate each other through a positive feedback mechanism, potentially mediated by cross-talk mechanisms that have been identified in other systems (*Akhshi et al., 2014*).

In combination with studies in other systems, our results suggest that mechanical regulation is a major avenue of crosstalk from the actin cytoskeleton to microtubule growth during neurite outgrowth. Consistent with this finding, it has also been shown that depolymerizing actin in growth cones spatially de-restricts microtubule polymerization, thus promoting neurite growth (*Bradke and Dotti, 1999*). A later study further found that ADF/cofilin, an actin severing enzyme, promotes neuritogenesis by disassembling the actin meshwork at the cortex, allowing for microtubule polymerization to promote a new protrusion (*Flynn et al., 2012*). In conjunction with these previous studies, our results argue that the actin cortex structurally inhibits microtubule extension, and that a stent-like broadening of the neurite shaft or other mechanisms to "loosen" the actin cortex, by actin waves or other processes, relieves this inhibition to enable more microtubules to extend forward.

In the context of neuron polarization, the stochastic nature of actin waves allows each neurite to repeatedly grow and retract thereby allowing the growth cone of each neurite to spatially explore its local environment to find external axon-promoting inputs (e.g. *Arimura and Kaibuchi, 2007*). When receiving sufficient input, signaling in one of the neurites is expected to be strengthened and the neurite is marked as the future axon. Such a specification of one neurite as the axon is known to subsequently inhibit axonal maturation of the other neurites and convert them to dendrites (*Esch et al., 1999*; *Ménager et al., 2004*; *Shelly et al., 2010*). Together with these previous considerations, our study argues that actin waves have a dual function. First, they are the drivers for neurite outgrowth by directing microtubule-based transport to deliver axon-promoting factors to the growth cones at the neurite tip. Second, they promote competition between neurites by generating large stochastic fluctuations that allow each neurite to sample "winning" and "losing" states as they explore their surroundings for the cues that will ultimately determine axon specification.

Our study does not directly address the role of actin waves in in vivo symmetry breaking and single axon formation. However, imaging of later stage polarized neurons in culture showed that actin waves still move through the nascent dendrites and the axon, but their effect on neurite outgrowth appears to be smaller (*Video 9*). This suggests that actin waves may become less important after polarization. With respect to their in vivo relevance, actin waves have been observed in hippocampal and cortical neurons in slice cultures (*Flynn et al., 2009*; *Katsuno et al., 2015*), although whether they promote microtubule polymerization and transport is more difficult to explore in this setting. Nevertheless, the results of our study suggest that actin waves could play an equally important role in neurite extension in vivo, to allow neurites to properly sense the polarized growth cues provided by the developing brain.

Together, our study provides a key missing link in our understanding of the axonal symmetry breaking process by demonstrating that the multi-polar state is characterized by actin waves driving microtubule-based transport, a link that we show is likely generated by an actin wave-triggered stent-like dilation of the neurite shaft. Our study further argues that the stochastic nature of actin waves creates the previously described stochastically fluctuating microtubule-based transport events during the multi-polar state. Thus actin waves may play a critical role in a stochastic search

mechanism that allows a set of neurites to explore the surrounding space, integrate external signaling cues, and measure relative input differences to select the a "winning" axon.

## Materials and methods

### Primary hippocampal neuron culture

Primary hippocampal neurons were cultured from Wistar rat E18 embryos. Hippocampi were removed from embryonic brains, placed in HBSS (Gibco, Life Technologies, NY) and incubated with 0.25% trypsin (Life Technologies) and 0.1% DNase (Roche Applied Science, Penzberg, Germany) for 15 min at 37°C. Hippocampi were then mechanically dissociated using gentle pipetting. Neuronal cells were either plated at 20,000–50,000 cell/cm$^2$ or electroporated using the Nuclefector Amaxa system (Lonza, Basal, Switzerland, Rat Neuron Nuclefector Kit, CN # VPG-1003) according to manufacturer's instructions then plated at 50,000 cell/cm$^2$. Cells were plated in MEM + Glutamax (Gibco) containing 3% glucose and 10% serum and then switched to standard culturing media (Neurobasal media (Gibco) with 1% Pen/Strep/Glut (Life Technologies), Glutamic acid (Life Technologies) and 2% B27 (Gibco) or 2% SMI (Stemcell Technologies, Vancouver, Canada)) 2–4 hr following plating. Cell were plated in LabTek II Chambered Coverglass chambers (Nunc, Rochester, New York), 96-well glass bottomed plates (In Vitro Scientific, Sunnyvale, CA) or 96-well plastic bottomed plates (Costar, Corning, New York) coated with poly-l-lysine (0.1 mg/mL, MW 30,000–70,000 Sigma Aldrich) or poly-l-lysine and laminin (2 µg/mL, Mouse Protein, Natural, Life Technologies).

### Drugs

Jasplakinolide (Santa Cruz Biotechnology, Dallas, Texas) was used at 10 nM (extracellular buffer) or 50 nM (CO2 independent media) depending on which media was used. Nocodazole (Sigma Aldrich, St. Louis, Missouri), was used at concentrations specified in the Figure Legends. Taxol (Abcam, Cambridge, MA) was used at 50 nM.

### Constructs

The Rac1 and Cdc42 FRET sensors were gifts from Michiyuki Matsuda and used the optimized backbone described in *Komatsu et al., 2011*. All other constructs used were cloned to express under a CAG promoter. Original F-tractin, EB1 (human), Synatophysin (rat), and CA-KIF5C (a.a. 1–560 of KIF5C (rat)) constructs were gifts from Michael Schell, Clare Waterman, Craig Garner, and Gary Banker respectively. mCerulean-PA-Rac1 was ordered from Addgene (Addgene #22030). Lyn-mCherry was used as a membrane marker, with the N-terminal Lyn sequence used described in (*Inoue et al., 2005*). The original GFP-KIF5C(1–560) vector was reassembled with Venus or Turquoise with the pCAGEN (Addgene #11160 [*Matsuda and Cepko, 2004*]) backbone using PCR assembly methods while maintaining the original linker sequence. For the remaining constructs, all final constructs were generated by Gateway cloning using the same methodology. First, a destination vector with the desired CAG promoter (for higher, more even expression in primary cells [*Qin et al., 2010*]) was generated. pCAGEN-DEST was constructed by taking the pCAGEN backbone and using the Gateway Vector Conversion System. Briefly, a Gateway cassette (Life Technologies) containing attR recombination sites with the ccdB gene and a chloramphenicol-resistance gene was inserted into the EcoRV restriction site contained within pCAGEN. Full-length YFP-EB1, F-tractin-mCherry (*Wollman and Meyer, 2012*), pCerulean-PA-Rac1, and Citrine-Synaptophysin was PCRed with the TOPO-compatible tag CACC at the 5' end and put into a pENTR vector using the pENTR/D-TOPO vector kit (Life Technologies). The resulted entry clones were then subject to an LR reaction (LR Clonase II Enyzme Mix, Life Technologies) with pCAGEN-DEST.

### Cell fixation and Immunofluorescence

Hippocampal neurons were fixed for 20 min in 4% paraformaldehyde and 4% sucrose in PBS. 2x fixation solution was added to native media. For IF, neurons were blocked for 1 hr in blocking/staining solution (3% Normal Goat Serum, 0.5% BSA, 0.2% TX-100 in PBS), incubated in primary antibody for one hour in blocking/staining solution and incubated in secondary antibody for one hour in blocking/staining solution, with standard washes. Primary antibodies: Neuronal Class III β-tubulin (TUJ1,

1:1000 dilution, Covance, Princeton, New Jersey). Secondary antibodies: Alexa-Fluor 488 (1:1000, Life Technologies). Dyes: Alex-Fluor 594 Phalloidin (1:400, Life Technologies).

## Microscopy

Live cell imaging was conducted on a Zeiss Axiovert 200 M inverted epifluorescent microscope (Zeiss, Oberkcochen, Germany) equipped with a Nipkow spinning disc confocal and 488, 514 and 594 nm lasers. Images were acquired on a CoolSnap HQ CCD camera (Photometrics, Tucson, Arizona) using 20x (0.75 NA), 63x water (1.2 NA), and 100x oil (1.4 NA) Zeiss objectives. Images were acquired with Micro-Manager (*Edelstein et al., 2010*) and processed using methods described below. For some experiments, live cell and fixed cell images were acquired on the ImageXpress Micro XLS Widefield High Content Screening System (Molecular Devices, Sunnyvale, CA) using 20x (0.45 or 0.75 NA) Nikon objectives. Photo-excitation studies were performed on a Leica SP8 scanning confocal microscope with a white-light laser and a 40x (1.3 NA) objective. Photo-excitation was performed by locally exciting a region of the neuron with 480 nm light on the white-light laser (10 rounds of 5 excitations, 10 sec between rounds, 10% power on the FRAP user interface). Structured illumination microscopy was performed on a GE/Applied Precision OMX V4 (GE Healthcare, Little Chalfont, UK) at the Neuroscience Microscopy Service at Stanford. All live cell imaging was performed at 37C in the absence of $CO_2$ unless otherwise noted. For imaging, $CO_2$ independent media (Gibco) or custom-made extracellular imaging buffer was used with added Pen/Strep/Glut (Life Technologies) and B27 (Gibco) or SMI (Stemcell Technologies) as a vitamin supplement.

## Image analysis

Image analysis was conducted using basic image analysis tools available in Fiji (*Schindelin et al., 2012*) or custom-written MATLAB scripts. *Total neurite intensity*: analysis was performed by generating a binary mask of the summed timelapse set of images, and assessing total intensity within the mask for each timepoint. *2D line scans*: analysis was conducted as follows 1) generate binary mask of neurite section of interest, 2) skeletonize mask, 3) select two endpoints on skeleton to generate "shortest path" aka a single path, 4) divide path into regular coordinates with desired spacing, 5) divide original binary mask into windows based on proximity to the closest coordinate. Using the window analysis, parameters such as mean signal intensity (from live cell probes or fixed cell staining), max signal intensity, neurite width (total number of pixels divided in a window divided by given spacing), puncta per window, and others can be calculated. In most cases, intensity measurements were determined by taking the mean or median of the top 20% of pixels in a given window. *Trace alignment for averaging data between independent waves*: traces were aligned to the half-maximum value of the front of the actin wave (oriented towards the growth cone). Half maximum values were determined by manual identification of the maximum and minimum actin signals on the front half of the actin curve. Traces were then computationally aligned. *EB1 puncta counting:* to count EB1 puncta, puncta were manually identified and computationally assigned to their corresponding window. *EB1 flow measurements*: to calculate flow, the number of EB1 puncta passing through a perpendicular plane of the neurite was counted by eye for 20 s. *2D kymographs*: to generate 2D kymographs, window analysis was used with coordinate spacing of 1–3 pixel(s). For each window, the maximum signal from the various sensors was measured. For kymographs in *Figure 6a* and *Figure 8c*, the "Multi Kymograph" feature of Fiji was used. *Mobile vesicle intensity measurement:* to obtain a measure of mobile vesicles, fast image series of 600 $m^{-1}$ s per image were gathered. Images were filtered using a tophat filter with a disc (2–3 pixel radius) to remove background signal. Negative values were then set to zero and the difference between subsequent images was taken in order to remove stationary particles. The resulting subtracted images from successive timepoints were added together and the summed image was subjected to the window analysis to produce line traces. *Single microtubule counting*: maximum intensity projection of SIM images were taken and the "Plot Profile" function used to take a line scan perpendicular to the neurite at desired locations in Fiji. Peaks in the line scan were identified by eye and counted as single microtubules. This method likely undercounts microtubules because of the high density of microtubules present in the neurite. *SIM images*: raw SIM data was processed on the API DeltaVision OMX softWoRx image processing software available in the Stanford Neuroscience Microscopy facility. Generation of maximum intensity projections and 3D reconstructions were performed in Fiji. *FRET analysis:* CFP and FRET images

were collected with a 20x (NA = 0.75) Nikon objective. Regions of interest were identified and subjected to a flat background subtraction (background calculated after removal of the object of interest), segmentation, and smoothing with a Gaussian filter before calculating FRET/CFP ratios. Custom written Matlab scripts used for 2D line scan analysis, mobile vesicle intensity measurements, and FRET analysis can be found at github.com/MeyerLab/AWinans_Elife_2016.

## Acknowledgements

Thanks to S Leal-Ortez and C Garner, M Matsuda, C Waterman, G Banker, and M Schell for constructs. Thanks to A Jaimovich, A Hayer, G Dey, S Cappell, H Yang, D Garbett, C Liu, and A Rana for comments and suggestions. Thanks to A Olsen and the Neuroscience Microscopy Service center for equipment use and training. This work was supported by RO1MH095087 and the Stanford Center for Systems Biology. AW was supported by the Stanford Biophysics Training Grant from the National Institutes of Health and the National Science Foundation Graduate Research Fellowship. SIM experiments were performed in the Stanford Neuroscience Microscopy Service, supported by NIH NS069375.

## Additional information

### Funding

| Funder | Grant reference number | Author |
|---|---|---|
| National Institutes of Health | RO1MH095087 | Tobias Meyer |
| National Science Foundation | Graduate Research Fellowship Program | Amy M Winans |
| National Institutes of Health | Stanford Biophysics Training Grant | Amy M Winans |
| National Institutes of Health | GM063702 | Tobias Meyer |

The funders had no role in study design, data collection and interpretation, or the decision to submit the work for publication.

### Author contributions

AMW, Conception and design, Acquisition of data, Analysis and interpretation of data, Drafting or revising the article; SRC, TM, Conception and design, Analysis and interpretation of data, Drafting or revising the article

### Author ORCIDs

Tobias Meyer, http://orcid.org/0000-0003-4339-3804

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
