## [Decision Letter]

Thank you for resubmitting your work entitled "Waves of actin and microtubule polymerization drive vesicle transport and neurite extension before axon formation" for further consideration at *eLife*. Your revised article has been favorably evaluated by Vivek Malhotra (Senior editor) and the original three reviewers, one of whom is a member of our Board of Reviewing Editors. The manuscript has been improved but there are some remaining issues that need to be addressed before acceptance, as outlined below. These revisions do not require new experiments.

The authors showed that latrunculin A treatment dilates the neurite to a similar extend as an actin wave. This is a really interesting finding. It suggests that the actin network restricts the protruding microtubules. If this were correct, then one would expect that latrunculin treatment would lead to microtubule protrusion and to axon growth. Hence, actin depolymerization would decouple the waves from axon growth, because the shaft widening and microtubule protrusion would be the important feature.

Please check Bradke and Dotti, Science, 1999. This appears exactly to be the case: Microtubules protrude into the distal area in the absence of actin dynamics upon pharmacological actin depolymerization. Similarly, please check Flynn et al., Neuron, 2012, where it is shown that cofilin renders the actin network dynamic and permissive for microtubule protrusion. In fact, the loss of cofilin can be restored by actin depolymerization. In principle, the data of the authors in conjunction with these papers may suggest that the waves formed is a way to make the actin cytoskeleton less restrictive for protruding microtubules. Please discuss these mentioned works accordingly as a possible action of how the waves could act to support axon growth. I don't think that these previous findings take anything away from the novelty of the beautiful findings presented here, but it will help the reader to put things in a very clear context.

[Editors’ note: a previous version of this study was rejected after peer review, but the authors submitted for reconsideration. The previous decision letter after peer review is shown below.]

Thank you for choosing to send your work entitled "Stochastic neurite extensions that precede axon formation are propelled by actin wave-driven microtubule-based transport" for consideration at *eLife*. Your full submission has been evaluated by Randy Schekman (Senior editor) and three peer reviewers, one of whom is a member of our Board of Reviewing Editors, and the decision was reached after discussions between the reviewers. Based on our discussions and the individual reviews below, we regret to inform you that your work will not be considered further for publication in *eLife*.

The reviewers agree that your observations are really interesting and exciting, and suggest a new paradigm for organization of microtubule-dependent traffic along the extending axon. However, the findings are largely descriptive, and the chain of causality is weak. It seems that quite a number of new experiments would be required to provide a convincing mechanism. Accordingly, we have decided to reject, but would be very interested in seeing a new version if you are able to develop the story further. Our full reviews are provided below.

Reviewer #1:

This paper relates actin waves, which travel anterogradely along neurites, to microtubules and vesicular movement. The studies were done with cultured hippocampal neurons, before an axon is permanently specified and when all neurites have MTs arranged with plus ends out. The authors find that CA-KIF5C moves to and fro prior to a wave, but the arrival of a wave causes a widening of the shaft and predominantly anterograde movement of CA-KIF5C. The CA-KIF5C and actin wave move together along the shaft, followed by vesicles and a wider shaft with increased microtubules. The authors repeat Ruthel and Banker's finding that MTs are needed for the actin waves, and also show that stabilizing F actin with Jasplakinolide inhibits the MT increase that occurs in the wave region. The results suggest positive feedback between the actin wave and MT processes.

1) The study is essentially correlative and it is not possible to say that the actin wave "drives" the MT-based transport. The Nocodazole result suggests that MTs actually drive the actin waves. The proposal that the increased F actin in the wave increases the volume, allowing more MT polymerization that is maintained as the wave passes through, is interesting, but basically untestable since there is no other way to create a dilation. Overall the results are interesting but rather descriptive. Potentially, localized inhibition of actin/MT dynamics partway along the shaft might allow cause and effect to be distinguished. However, at the present time, it is speculative to assert that actin waves "propel" neurite extensions or "we discovered.… a mechanical control system whereby the actin waves widen the neurite shaft and create the space needed for more microtubules to polymerize".

2) Another concern is that actin waves have only been seen in vitro, with cultured neurons. Ruthel and Banker 1998 show what may be an actin wave progressing down the axon of a single hippocampal neuron in slice culture. However, to my knowledge, waves have not been noted in the many publications in which neurons were labeled by in utero electroporation and then imaged in slice culture. As noted in Barnes and Polleux, 2009, in vivo pyramidal neurons inherit their axon-dendrite polarity from the trailing and leading processes during migration. The hippocampal culture system is essentially a "repolarization" of cells that were polarized in vivo before they were dissociated for culture. Studying symmetry breaking using the hippocampal culture system is well established, but the authors should be explicit about the artificiality of the system.

3) The mechanical control system is not clear. Does the increased MT extension (or initiation) in the wave result from the increased cross-section area or from increased rate of extension of individual MTs per unit volume? The super-resolution microscopy suggests that the microtubule number is greatly increased behind the wave. This means there are more MT ends available for extension. Perhaps EM would help show whether more MTs are being seeded in the wave, or whether the dilation is creating volume for more tubulin dimers to recruit to the growing plus ends.

Reviewer #2:

This manuscript reports live imaging of actin waves in multipolar hippocampal neurons in culture. The waves are associated with what appears to be increased lamellipodial activity along the process shaft, and an increase in process diameter. Growing microtubule ends, vesicular cargo, and kinesin motor domain are all enriched in the wave and may follow it as it moves outward along the neurite.

The strengths of this study are that the actin wave phenomenon is very interesting, and the efforts at reliable quantitative characterization are impressive. Determining wave function within an initial study would be a lot to ask, but some further clarity on the limits of the study is needed. In particular, the physiological purpose of the waves is incompletely understood from the current data. Also, there is little evidence other than codistribution with CA-KIF5C and synaptophysin vesicles for a role in axon specification, as implied in the title and text. As a practical issue, it has remained unresolved how reliably the CA-KIF5C motor domain alone can locate the putative nascent axonal process. The current study seems to push the question back to the actin wave, but it remains unclear whether the CA-KIF5C construct is simply carried forward as part of the increased cytoplasmic volume, whether it is enriched solely as a result of the increased MT number, or whether it localizes specifically to MT plus ends, not a known form of behavior for this kinesin. Another issue is the limited data indicating that actin actually leads the way in this behavior. Modifications and additions to the text and data are recommended to clarify these and other important issues and to improve the presentation of the data.

1) Based on the frequency per neurite and the velocity of the actin waves, they would be expected to appear very commonly in unpolarized neurons, perhaps as lateral lamellipodial protrusions in at least one or more neurites in general, and not just in this study. Are they detectable in published images, and how commonly? Or might they be a product of specific culture conditions used here?

2) The function of the waves in neuronal differentiation remains largely unaddressed and is a critical issue for understanding the physiological importance of these structures. Neurons should be monitored for an additional day to correlate actin wave activity with longer-term neurite extension and axonogenesis. It seems as though there may be no direct link between the waves and axon formation, in which case the waves may remain for the moment a curiosity, albeit a remarkable one.

3) This relates to another issue, the localization of CA-KIF5C to neurite tips. This is reported to correlate strongly with axon identity, yet CA-KIF5C appears to be present in most or all neurites in the current study. What are the implications for CA-KIFC as a pre-axon marker?

4) In the first paragraph of the subsection “Actin waves are linked to anterograde Synaptophysin-mediated vesicular transport during the multi-polar phase of axon formation“: The authors report actin wave velocities of 2-3 μm/min in Figure 1, but less than this < 2 μm/min in Figure 9 and Figure 5 controls.

5) Figure 2: This graph represents an important form of analysis, in showing the temporal sequence of actin vs. CA-KIF5C at a point within the neurite. Surprisingly, this is the only such analysis shown, but serves as the basis for a major claim (see summary diagram and Title) that actin leads the way. More of this type of analysis comparing actin with other subcellular structures should be provided.

6) In the first paragraph of the subsection “Actin waves are linked to anterograde Synaptophysin-mediated vesicular transport during the multi-polar phase of axon formation “and Figure 1: The authors claim that "arriving actin waves.…[were]… in most cases followed by neurite extension." However, only 8/16 showed some degree of neurite extension.

7) Figure 1: The authors relied on only the moving vesicles to plot the curve on the right. How does a plot of total vesicles appear (moving and non-moving)? This issue gets at whether bulk cytoplasm is moving along with actin, or, instead, whether microtubule-based transport is stimulated within the region of the wave.

8) What is meaning of stalled EB1, which is reported to associate as short lines at the growing plus ends of microtubules?

9) What is the three-dimensional nature of the F-tractin peak. Are we seeing anterograde transport of F-actin filaments, a bulge of swollen cytoplasm, or a combination of lamellipodia and surface ruffles?

10) Figure 5: The superresolution image showing MT density in the wave is useful, but a comparison with MT density ahead of and behind the wave would make more sense to include.

Reviewer #3:

The manuscript "Stochastic neurite extensions that precede axon formation are propelled by actin wave-driven microtubule-based transport" by Winans et al. report that actin waves, which have been formerly shown to precede neuronal polarization, enhances traffic into the future axon. Moreover, the authors show that actin waves enhance microtubule polymerization. The authors further reveal that the effect on microtubule polymerization is transient, which open up the possibility to form a stochastic model of how neuronal polarization takes place.

In general, the work is very fascinating and gives a very interesting perspective of how neuronal polarization can occur. While their exciting work has the potential to become a fine *eLife* paper, the authors need to perform some experiments to put their idea on a more solid basis. All of the experiments are relatively straightforward and should be all done.

1) The authors show that kinesin-1 mediated transport is enhanced after waves. What happens when kinesin 1 activity is abrogated? Do the authors still see traffic into the axon? Do waves actually still form without kinesin 1 activity?

2) The authors argue that the actin waves dilate the neurites to enable enhanced microtubule polymerization and microtubule based transport. Could the actin waves enable this process by dynamizing the cortical actin. Consistent with this possibility the authors show that jasplakinolide prevents these processes. The authors should try the converse experiment with cytochalasin D or latrcunculin. If their hypothesis is true then cytoD treatment (probably local treatment) should enable dilation and enhance microtubule polymerization. Please check also Bradke and Dotti, Science, 1999, where preferential trafficking and axon formation has been reported after local actin depolymerization.

3) What happens if the cultures are treated with low doses of Taxol (3 nM) that enables microtubule polymerization (Witte et al., JCB, 2008)? How does it affect microtubule polymerization and dilation of the neurite? Is there an effect on wave formation?

4) Figure 5: According to the Methods part, the authors fixed the cells and subsequently extracted them. Therefore, soluble tubulin is left in these staining. The authors should perform a simultaneous fixation-extraction so that tubulin dimers are gone for the actual staining procedure.

5) The authors should mention in the paper that it had been previously shown that enhanced membrane traffic precedes axon formation (Bradke and Dotti, Neuron, 1997). I don't think that this info will take anything away from their exciting findings.

---

## [Author Response]

*The authors showed that latrunculin A treatment dilates the neurite to a similar extend as an actin wave. This is a really interesting finding. It suggests that the actin network restricts the protruding microtubules. If this were correct, then one would expect that latrunculin treatment would lead to microtubule protrusion and to axon growth. Hence, actin depolymerization would decouple the waves from axon growth, because the shaft widening and microtubule protrusion would be the important feature. Please check Bradke and Dotti, Science, 1999. This appears exactly to be the case: Microtubules protrude into the distal area in the absence of actin dynamics upon pharmacological actin depolymerization. Similarly, please check Flynn* et al.*, Neuron, 2012, where it is shown that cofilin renders the actin network dynamic and permissive for microtubule protrusion. In fact, the loss of cofilin can be restored by actin depolymerization. In principle, the data of the authors in conjunction with these papers may suggest that the waves formed is a way to make the actin cytoskeleton less restrictive for protruding microtubules. Please discuss these mentioned works accordingly as a possible action of how the waves could act to support axon growth. I don't think that these previous findings take anything away from the novelty of the beautiful findings presented here, but it will help the reader to put things in a very clear context.*

This is an excellent point, and we now discuss our findings in the context of these previous papers in the Discussion.

[Editors’ note: the author responses to the previous round of peer review follow.]

*Reviewer #1: 1) The study is essentially correlative and it is not possible to say that the actin wave "drives" the MT-based transport. The Nocodazole result suggests that MTs actually drive the actin waves. The proposal that the increased F actin in the wave increases the volume, allowing more MT polymerization that is maintained as the wave passes through, is interesting, but basically untestable since there is no other way to create a dilation. Overall the results are interesting but rather descriptive. Potentially, localized inhibition of actin/MT dynamics partway along the shaft might allow cause and effect to be distinguished. However, at the present time, it is speculative to assert that actin waves "propel" neurite extensions or "we discovered*.…

*a mechanical control system whereby the actin waves widen the neurite shaft and create the space needed for more microtubules to polymerize".*

Reviewer 1 critiques the causality links in the paper. In the previous version of the paper, we emphasized a linear causality link whereby actin waves increase microtubule polymerization which leads to increased microtubule-based transport. Reviewer 1 argues that our hypothesis concerning a mechanical link between actin waves and microtubules is not sufficiently supported, as is the link between actin waves and neurite outgrowth. Although we show that stalling actin waves with the drug Jasplakinolide prevents the forward progress of the wave of polymerizing microtubules and CA-KIF5C, Reviewer 1 further argues that the Nocodazole data, whereby inhibiting microtubule polymerization leads to the dissipation of actin waves, shows a co-dependence of polymerizing actin and microtubules.

We also agree that it is likely that actin waves and polymerizing microtubules are co-dependent on one another, although this model was not emphasized in the previous version of the manuscript. In fact, there are multiple studies examining the positive feedback between actin and microtubule polymerization, as is mentioned in our previous Discussion. To address the concerns of Reviewer 1 we have added emphasis to a model of co-dependence in the new version of the manuscript, and have also added data showing that actin waves are affected by the addition of Taxol, a pharmacological microtubule stabilizer (as suggested by Reviewer 3). This compliments the Nocodazole data and further supports a model of co-dependence.

Nonetheless, Reviewer 1 makes a valid point that the links between actin waves, the increase in neurite volume, and microtubule polymerization, required more supporting evidence. To address these concerns, we have used a photo-activatable Rac1 construct to generate actin waves de novoand probe the causality chain starting from the actin wave showing increased microtubule polymerization. We also performed further experiments to support our argument concerning a mechanical role of actin waves in microtubule polymerization.

*2) Another concern is that actin waves have only been seen* in vitro*, with cultured neurons. Ruthel and Banker 1998 show what may be an actin wave progressing down the axon of a single hippocampal neuron in slice culture. However, to my knowledge, waves have not been noted in the many publications in which neurons were labeled by in utero electroporation and then imaged in slice culture. As noted in Barnes and Polleux, 2009,* in vivo

*pyramidal neurons inherit their axon-dendrite polarity from the trailing and leading processes during migration. The hippocampal culture system is essentially a "repolarization" of cells that were polarized in vivo before they were dissociated for culture. Studying symmetry breaking using the hippocampal culture system is well established, but the authors should be explicit about the artificiality of the system.*

Reviewer 1 brings up a concern about the in vivo significance of the actin waves, a concern which is shared by Reviewer 2. It is true that actin waves have been overwhelmingly examined in only an in vitrocontext. There are two known observations of actin waves in slice cultures; the first by Flynn et al., Develop Neurobiol., 2009, and the second in the recently-published Katsuno et al., Cell Reports, 2015. During the revision process, we reached out to several zebrafish developmental biologists to see if they have observed actin waves. Unfortunately, they had not, although in all cases the developmental window in which they imaged and the temporal resolution of imaging was not suited to observe actin waves, if they did exist. In in vitrocultures, actin waves are most frequent in young cultures (DIV1-2) in pre-polarized neurons. We also image a minimum of once every five minutes to capture the movement of the actin waves. It is very possible actin waves have not been observed more widely in vivo because the imaging parameters have not been suited to capturing the movement of actin waves.

Reviewer 1 also notes that pyramidal neurons do inherit their axon-dendrite polarity from their trailing and leading processes during polarization. The review cited (Barnes and Polleux, 2009) also describes a multi-polar stage before the establishment of the leading and trailing processes; a stage that in the retinal ganglion of zebrafish (Randlett et al., Neuron, 2011) bears the characteristic switching of CA-KIF5C in between neurites. Because of these data, we do think that the fluctuating neurite outgrowth and microtubule-based transport observed during the multi-polar phase is important to ensuring the specification of the leading and trailing process in the correct orientation, which leads to axon specification. However, we have modified the manuscript to better reflect the limitations of the in vitrosystem we use, as suggested by the reviewers.

3) The mechanical control system is not clear. Does the increased MT extension (or initiation) in the wave result from the increased cross-section area or from increased rate of extension of individual MTs per unit volume? The super-resolution microscopy suggests that the microtubule number is greatly increased behind the wave. This means there are more MT ends available for extension. Perhaps EM would help show whether more MTs are being seeded in the wave, or whether the dilation is creating volume for more tubulin dimers to recruit to the growing plus ends.

Reviewer 1 points out that the data on the mechanical control system is incomplete and suggests some excellent experiments to provide more evidence for this model. We have added this and other experiments to help support our model, although we would like to emphasis that this is not necessarily the only means that actin waves can increase microtubule polymerization, but the simplest one that fits the data we have gathered.

To look at whether the increase in the number of polymerizing microtubule plus-ends was a proportional increase to width, or increased per unit area in an actin wave, we counted the flow of EB1 puncta (the number of puncta to pass through a plane in a neurite in a set amount of time – this is not per unit width) in different widths of smooth neurite shafts, and in the shafts of neurites bearing actin waves (Figure 5). We found that there was a roughly linear relationship between neurite width and EB1 puncta flow. (Note that this measurement is of a single z-plane and does not count the entire volume of the neurite). We further found that the flow per neurite width in actin waves fell within the distribution of flow measurements in smooth, control neurites. This data supports a positive correlation between neurite width and the number of polymerization microtubules, and affirms that the increase in EB1 puncta in actin waves is consistent with the increase in width.

We have further showed that at a bottleneck, where the neurite rapidly narrows, we see the disappearance of EB1 puncta (Figure 5, right), similar to what you would see at the leading edge of a cell, or at a growth cone (Figure 5, left). This data also supports a model where microtubules prefer to polymerize into open space, and will stop polymerizing if they reach a steric restriction.

Finally, we added additional statistics to the bottleneck experiments performed in the previous version of the paper, re-affirming that over time, polymerizing microtubules upstream of a bottleneck will not all successfully pass through the bottleneck (Figure 5). We believe this data further strengthens the argument that microtubule polymerization is affected by steric constraints, rather than itself solely dictating the width of the neurite.

Reviewer #2: The strengths of this study are that the actin wave phenomenon is very interesting, and the efforts at reliable quantitative characterization are impressive. Determining wave function within an initial study would be a lot to ask, but some further clarity on the limits of the study is needed. In particular, the physiological purpose of the waves is incompletely understood from the current data. Also, there is little evidence other than codistribution with CA-KIF5C and synaptophysin vesicles for a role in axon specification, as implied in the title and text. As a practical issue, it has remained unresolved how reliably the CA-KIF5C motor domain alone can locate the putative nascent axonal process. The current study seems to push the question back to the actin wave, but it remains unclear whether the CA-KIF5C construct is simply carried forward as part of the increased cytoplasmic volume, whether it is enriched solely as a result of the increased MT number, or whether it localizes specifically to MT plus ends, not a known form of behavior for this kinesin. Another issue is the limited data indicating that actin actually leads the way in this behavior. Modifications and additions to the text and data are recommended to clarify these and other important issues and to improve the presentation of the data.

The concern that CA-KIF5C is simply reporting on cytoplasmic volume is addressed in Reviewer 2’s point 5.

1) Based on the frequency per neurite and the velocity of the actin waves, they would be expected to appear very commonly in unpolarized neurons, perhaps as lateral lamellipodial protrusions in at least one or more neurites in general, and not just in this study. Are they detectable in published images, and how commonly? Or might they be a product of specific culture conditions used here?

See response to Reviewer 1 point 2.

2) The function of the waves in neuronal differentiation remains largely unaddressed and is a critical issue for understanding the physiological importance of these structures. Neurons should be monitored for an additional day to correlate actin wave activity with longer-term neurite extension and axonogenesis. It seems as though there may be no direct link between the waves and axon formation, in which case the waves may remain for the moment a curiosity, albeit a remarkable one.

Reviewer 2 brings up an excellent point concerning the role of actin waves during the precise step of symmetry breaking and following axon specification. Flynn et al. (2009) previously observed an upregulation of the number of actin waves in the neurite that becomes the axon. In post-polarized neurons, we still observe the production of actin waves in the shorter, non-axonal processes.

However, these actin waves have a greatly diminished effect on neurite outgrowth (Video 9). We speculate that at this point, other mechanisms have been established to biaspolarized transport to the nascent axon, and the role of actin waves in controlling microtubule-based transport is diminished. We furthermore observe that once axons grow long, actin waves are less likely to travel all the way to the axonal tip. This also correlates with more observed neurite outgrowth independent of actin waves in more mature neurons. Although this matter could and should be further clarified, we respectfully argue that this is beyond the scope of this particular story.

To interpret what our data suggests in the light of axon specification, we believe that actin waves link neurite outgrowth to microtubule-based transport in order to help neurites grow out and explore the extracellular space for growth cues, while the neurite can also receive internal pro- growth cues. This is only one part of the kinetics of axon specification though. The maintenance of growth cues at the neurite tip might not just depend on actin waves, but also the presence of external growth cues that can prevent growth cone collapse and retrograde movement of the components in the growth cone.

3) This relates to another issue, the localization of CA-KIF5C to neurite tips. This is reported to correlate strongly with axon identity, yet CA-KIF5C appears to be present in most or all neurites in the current study. What are the implications for CA-KIFC as a pre-axon marker?

We examine CA-KIF5C localization in pre-polarized neurons, when the axon has yet to be determined. Multiple studies have shown that CA-KIF5C localizes to a single neurite or subset of neurites in multi-polar neurons in a dynamic fashion (Jacobson et al., Neuron, 2006; Toriyama et al., Mole Sys Biol., 2010, Randlett et al., Neuron, 2011). We use a virtually identical construct to that used by Banker et al. (Neuron, 2006), with a change in the fluorescent protein that preserves the original linker. Cloning is described in the Materials and methods. CA-KIF5C localization is hugely dynamic. The fluorescently-tagged motor protein will move in and out of neurites at a timescale on the order of tens of minutes. It can also dwell in the growth cones for variable amounts of time. Localization of CA-KIF5C at any given time will be determined by the balance of the anterograde and retrograde movement of the protein. Given our finding that the CA-KIF5C travels with the actin wave and is thus linked with neurite outgrowth, it appears that CA-KIF5C presence in the neurite coincides with a neurite’s attempt to grow out and become the axon. Its movement out of the neurite may occur when the signals traditionally responsible for axon specification are not strong enough to maintain CA-KIF5C and other possible pro-axon factors at the growth cone. During the multi-polar phase, each neurite has the inherent capability to become an axon, and thus most or all neurites will grow and receive growth promoting factors. Due to insufficient growth cues (both external and internal) however, most neurites will be unable to permanently specify as axons.

*4) In the first paragraph of the subsection “Actin waves are linked to anterograde Synaptophysin-mediated vesicular transport during the multi-polar phase of axon formation“: The authors report actin wave velocities of 2-3 μm/min in Figure 1, but less than this < 2 μm/min in Figure*

3E & 5E controls.

The speed of actin waves varies moderately from day-to-day and from neurite-to-neurite, perhaps influenced by imaging and culturing conditions. More photo-toxic imaging conditions can affect speed and possibly frequency; where the actin wave is measured in the neurite can also affect the measured speed. The speeds measured still all fall within an expected distribution. Importantly, controls are always assessed in parallel to perturbations performed in similar conditions.

5) Figure 2: This graph represents an important form of analysis, in showing the temporal sequence of actin vs. CA-KIF5C at a point within the neurite. Surprisingly, this is the only such analysis shown, but serves as the basis for a major claim (see summary diagram and Title) that actin leads the way. More of this type of analysis comparing actin with other subcellular structures should be provided.

To address this point, we first conducted a more precise analysis of the delay between actin and CA-KIF5C entry into the neurite by measuring the time delay (if any) between the increase in actin signal and the increase in CA-KIF5C (Figure 9). The re-analyzed data confirmed that an increase in actin signal often (although not always) precedes the increase in CA-KIF5C. This

observation is in agreement with our new photo-activation results that show that de-novo generated Rac1 activity is sufficient to initiate actin waves and increase microtubule polymerization. However, it is important to point out that we are simply measuring the first minutes of actin wave entry into the neurite. Once a burst of CA-KIF5C has entered the neurite, it rapidly catches up with the actin wave. We therefore have worked to emphasize in the text that actin and microtubules do work together to advance the wave of actin, polymerizing microtubules, and microtubule-based transport.

To further address the concern that the movement of CA-KIF5C is mirroring that of cytoplasm, we monitored the ratio of CA-KIF5C-Citrine to cytoplasmic Turquoise in the tips of neurites over time. We can see the ratio of these signals increase and decrease as CA-KIF5C moves in and out of the neurites, suggesting that the CA-KIF5C increases are distinct from increases in volume. Moreover, the addition of Nocodazole causes the ratio to become constant over time, affirming a role in polymerizing microtubules in the movement of CA-KIF5C. Reviewer 2 notes that it appears that CA-KIF5C may be moving with microtubule plus ends. An interpretation of our data could support this hypothesis, which has been suggested by other papers (Vale et al., J Biol Chem, 1994; Nakata et al., J Cell Biol., 2011). Of note, if there is a cytoplasmic fraction of CA-KIF5C that contributes to some of our intensity traces, we believe this ratiometric analysis supports the validity of the entry event analysis.

*6) In the first paragraph of the subsection “Actin waves are linked to anterograde Synaptophysin-mediated vesicular transport during the multi-polar phase of axon formation “and Figure 1: The authors claim that "arriving actin waves*.…*[were]*…

*in most cases followed by neurite extension." However, only 8/16 showed some degree of neurite extension.*

Addressed in Reviewer 1’s point 1.

*7) Figure 1: The authors relied on only the moving vesicles to plot the curve on the right. How does a plot of total vesicles appear (moving and non-moving)? This issue gets at whether bulk cytoplasm is moving along with actin, or, instead, whether microtubule-based transport is stimulated within the region of the wave.*

Following Reviewer 2’s suggestion, we examined the motile and non-motile Synaptophysin-positive vesicles in actin waves. Our movies indicate that a substantial portion of the vesicles are non-motile and fixed in place, not subject to Brownian motion (Video 6). This suggests that these vesicles are either attached to microtubules (or possibly actin) and are not simply cytoplasmic.

8) What is meaning of stalled EB1, which is reported to associate as short lines at the growing plus ends of microtubules?

Our writing was unclear when describing the result of the Jasplakinolide experiment (Figure 4). The EB1 puncta are still moving in a stereotypical fashion after addition of the Jasplakinolide within the neurite and the actin wave, meaning that microtubule polymerization is still taking place within the neurite and actin wave. However, the wave of EB1 puncta (or the wave of

polymerizing microtubules) largely disappears once it hits the bottleneck. We think this is another example of polymerizing microtubules being sterically hindered from moving past a bottleneck.

9) What is the three-dimensional nature of the F-tractin peak. Are we seeing anterograde transport of F-actin filaments, a bulge of swollen cytoplasm, or a combination of lamellipodia and surface ruffles?

To answer these questions, we performed additional super-resolution SIM experiments using phalloidin-stained neurons. 3D reconstructions of the actin waves (Figure 1) show a widened shaft with lamellipodia jutting out in a perpendicular manner. The lamellipodia structures stay mainly within the plane of the glass surface, with some dimension in 3D.

10) Figure 5: The superresolution image showing MT density in the wave is useful, but a comparison with MT density ahead of and behind the wave would make more sense to include.

An excellent point and an oversight on our part. New images were introduced to better illustrate the MT density before and after the actin wave. We also performed statistical analysis on number of microtubules cables before and after the actin waves (Figure 2—figure supplement 2) to reinforce the statistics taken from the low-resolution immunofluorescence data.

Reviewer #3: In general, the work is very fascinating and gives a very interesting perspective of how neuronal polarization can occur. While their exciting work has the potential to become a fine eLife paper, the authors need to perform some experiments to put their idea on a more solid basis. All of the experiments are relatively straightforward and should be all done. 1) The authors show that kinesin-1 mediated transport is enhanced after waves. What happens when kinesin 1 activity is abrogated? Do the authors still see traffic into the axon? Do waves actually still form without kinesin 1 activity?

Several previous studies have examined the effect of Kinesin 1 knock-down on axon formation and trafficking. Ferreira et al. (JCB, 1992) showed that knock-down of Kinesin 1 caused disregulation of the trafficking of GAP-43 and synapsin, and also caused shorter neurites.

Konishi and Setou (Nature Neuroscience, 2009) showed that knock-down of Kinesin 1 caused polarity defects in neurons (more multi-polar neurons) and shorter axons, and the introduction of a mutant Kinesin-1 causes erroneous distribution of Piccolo, a pre-synaptic protein. Knock-down of Kinesin-1 also causes an aberrant localization of CRMP2, a protein important for axonal development (Kimura et al., J. of Neurochemistry, 2005). These results suggest that Kinesin-1 does play a role in trafficking and polarity.

As per Reviewer 3’s suggestion, we also attempted to examine the effect of Kinesin-1 knock down on actin waves. We first attempted to knock down all three known isoforms of Kinesin-1, but were unable to achieve more than mild knock down of each isoform (assessed via qPCR). We then decided to do a single knockdown of KIF5C, an isoform highly expressed in the brain, and examine the effect of the single knock down on actin waves formation. After achieving ~65% knock down (assessed via qPCR), we found no apparent affect on axon formation, assessed at DIV3. This differs from published results (Konishi and Setou, 2009), likely because the paper cited was able to knock down all three isoforms. We also observed no affect on actin wave formation during a co-culture experiment of control and KIF5C knock-down, assessed on DIV2. siRNA’s were introduced before plating on DIV0. Moving forward, it is unclear whether the lack of phenotypes observed during knock-down is biologically significant, or a result of insufficient knock-down.

2) The authors argue that the actin waves dilate the neurites to enable enhanced microtubule polymerization and microtubule based transport. Could the actin waves enable this process by dynamizing the cortical actin. Consistent with this possibility the authors show that jasplakinolide prevents these processes. The authors should try the converse experiment with cytochalasin D or latrcunculin. If their hypothesis is true then cytoD treatment (probably local treatment) should enable dilation and enhance microtubule polymerization. Please check also Bradke and Dotti, Science, 1999, where preferential trafficking and axon formation has been reported after local actin depolymerization.

Per the reviewer’s suggestion, we observed what happened to actin waves and neurites in the presence of Latrunculin A. Interestingly, although LatA treatment did not appear to affect smooth neurites, we found that actin waves, or regions rich in f-actin, would dissolve (as expected) and often (though not always), caused neurite widening in various sections of the neurite shaft. In Figure 5—figure supplement 1, three examples are noted. In a, the neurite in front of the former actin wide widens; in b, the entire neurite widens; in c, the base of the neurite widens.

This may be an example of the actin cortex “softening” or dynamizing, allowing EB1 puncta to pass through, although we would be careful not to over-interpret the data.

3) What happens if the cultures are treated with low doses of Taxol (3 nM) that enables microtubule polymerization (Witte et al., JCB, 2008)? How does it affect microtubule polymerization and dilation of the neurite? Is there an effect on wave formation?

This is an excellent suggestion and a good way to further probe the connection between actin and microtubules. In response, we treated F-tractin-mCherry expressing neurons with increasing doses of Taxol (5 – 50 nM) and observed effects on actin waves. Although we did not see any obvious effects on actin waves at low doses within the several hours of exposure observed, high doses of Taxol disrupted the processive nature of actin waves (Figure 7). Bursts of actin polymerization were still observed, although the bursts could originate in different location in the neurite, rather than the cell body, and lacked the anterograde processive behavior of stereotypical actin waves. EB1 puncta measurements in the presence of a high dose of Taxol show that microtubule polymerization is diminished at high doses (Figure 7—figure supplement 1), so it remains unclear whether the effect of Taxol on actin waves is from the stabilization of microtubules, or from its affects on microtubules polymerization.

*4) Figure 5: According to the Methods part, the authors fixed the cells and subsequently extracted them. Therefore, soluble tubulin is left in these staining. The authors should perform a simultaneous fixation-extraction so that tubulin dimers are gone for the actual staining procedure.*

To answer, we attempted several types of fixation-extraction experiments, but were unable to fully extract a cytosolic fluorescent protein control. We decided to instead acquire additional SIM measurements in order to count single MT cables. These measurements would be independent of the signal from the tubulin dimers, and thus still address the referee’s comment. Our data showed that a greater number of microtubule cables were present in and behind the actin wave (Figure 2—figure supplement 2). This enrichment ratio was also statistically greater than the control ratio in Figure 2. Given that the presence of dimers would artificially increase both ratios, (experimental and the control ratio), we are satisfied that the single microtubule counts confirm an enrichment of microtubules behind the actin wave greater than would be caused by natural widening of neurites as you approach the cell body.

*5) The authors should mention in the paper that it had been previously shown that enhanced membrane traffic precedes axon formation (Bradke and Dotti, Neuron, 1997). I don't think that this info will take anything away from their exciting findings.*

An omission on our part that has been corrected.